# The molecular dynamics of subdistal appendages in multi-ciliated cells

Hyunchul Ryu[1,2,4], Haeryung Lee [1,4], Jiyeon Lee[1], Hyuna Noh[1], Miram Shin[1], Vijay Kumar[3], Sejeong Hong[1], Jaebong Kim[3] & Soochul Park [1✉]

The motile cilia of ependymal cells coordinate their beats to facilitate a forceful and directed flow of cerebrospinal fluid (CSF). Each cilium originates from a basal body with a basal foot protruding from one side. A uniform alignment of these basal feet is crucial for the coordination of ciliary beating. The process by which the basal foot originates from subdistal appendages of the basal body, however, is unresolved. Here, we show FGFR1 Oncogene Partner (FOP) is a useful marker for delineating the transformation of a circular, unpolarized subdistal appendage into a polarized structure with a basal foot. Ankyrin repeat and SAM domain-containing protein 1A (ANKS1A) interacts with FOP to assemble region I of the basal foot. Importantly, disruption of ANKS1A reduces the size of region I. This produces an unstable basal foot, which disrupts rotational polarity and the coordinated beating of cilia in young adult mice. ANKS1A deficiency also leads to severe degeneration of the basal foot in aged mice and the detachment of cilia from their basal bodies. This role of ANKS1A in the polarization of the basal foot is evolutionarily conserved in vertebrates. Thus, ANKS1A regulates FOP to build and maintain the polarity of subdistal appendages.

[1] Department of Biological Sciences, Sookmyung Women's University, Seoul 04310, Korea. [2] Department of Life Science, University of Seoul, Seoul 02504, Korea. [3] Department of Biochemistry, Institute of Cell Differentiation and Aging, College of Medicine, Hallym University, Chuncheon 24252, Korea. [4] These authors contributed equally: Hyunchul Ryu, Haeryung Lee. ✉email: scpark@sookmyung.ac.kr

Multi-ciliated cells (MCCs) are terminally differentiated epithelial cells that form the interface between the extracellular milieu and underlying tissues[1–4]. These specialized cell types are present mostly in vertebrates rather than in invertebrates. Each MCC contains several dozen or even hundreds of motile cilia depending on the cell type. For motile cilia, their coordinated beating generates a forceful and directed flow of fluid over the epithelial membrane[5–7]. Ciliary motility is critical for normal function in various tissues, including the circulation of cerebrospinal fluid (CSF) through brain ventricles, clearance of protective mucus from airways, and transport of eggs through oviducts[8–13]. But it remains unclear how MCCs orient with one another and coordinate ciliary beating at the molecular level. Motile cilia in brain ventricles must beat constantly, leading to significant shear stress throughout their lifespan. These MCCs may represent an ideal model for studying how the machinery of coordinated beating is set up and maintained.

In brain ventricles, the CSF delivers growth factors, maintains proper ion concentrations, and removes various metabolic byproducts[3,4]. Thus CSF flow and drainage must proceed continuously for proper brain homeostasis. Multi-ciliated ependymal cells, herein referred to as E1 cells, are the major cell type lining the ventricles. E1 cells have approximately 50 motile cilia per cell and play a key role in facilitating the directed flow of CSF through the ventricular system. To power-directed CSF flow, E1 cells must maintain a planar organization of their cilia[14,15]. In particular, the rotational organization of their basal bodies (BBs) is key to the coordinated beating of motile cilia[5,6,16,17]. Defects in the coordinated ciliary beating of E1 cells leads to abnormal CSF accumulation and hydrocephalus, as demonstrated in rodents.

For E1 cells and for other MCCs, the basal foot (BF) is crucial for coordinated ciliary beating[3,6,7]. The BF is an electron-dense conical structure that protrudes unilaterally from the BB barrel[6,10]. The BF connects the BB to the microtubule (MT) cytoskeleton, organizing neighboring BBs and embedding them a cytoskeletal network[18]. Importantly, the orientation of the BF points in the direction of CSF flow, thereby conferring a one-sided planar bias to the BB[3,6,7]. As it is with MCC in other tissues, the unidirectional alignment of all the basal feet on each E1 cell is referred to as the cell's rotational polarity. Neighboring E1 cells normally show parallel alignment of their basal feet, which is referred to as tissue-level rotational polarity. Not only is the disruption of rotational polarity associated with hydrocephalus[6,15] but has also been implicated in other diseases like primary ciliary dyskinesia in the respiratory tract[10,17,19]. E1 cells also display asymmetric ("off-center") positioning of their clustered BBs at the cell and tissue levels, which is referred to as "translational polarity"[3,4]. MCCs in the mouse trachea or frog epidermis do not show translational polarity, so it remains unclear whether the translational polarity in mature E1 cells contributes to coordinated ciliary beating.

In contrast to MCCs, which have only a single BF per BB, cells with primary cilia have multiple basal feet[20]. This may allude to the lack of motility of primary cilia. The basal feet in cells with primary cilia originate from subdistal appendages (SDAs), which are anchored to MTs and position the BBs. It should be noted that the nomenclature of basal feet versus SDAs is not clearly defined for cells with primary cilia[21–24]. Although the basal feet in motile cilia and primary cilia share parts of their molecular architecture, some of their proteins appear in different locations. For example, while outer dense fiber 2 (ODF2), which is an essential scaffold for initiating the assembly of basal feet/SDAs, appears near the centriolar wall along the transverse axis of both BF types, centriolin (CNTRL) appears near ODF2 in primary cilia but not in motile cilia. In motile cilia, CNTRL appears further away in the region where the BF tapers[20,25,26]. FGFR1 oncogene

partner (FOP) is also localized to the basal feet/SDAs of primary cilia[27], whereas its precise localization in the BB of motile cilia is unknown.

Ankyrin repeat and SAM domain-containing protein 1A (ANKS1A), which forms the ANKS1 family together with ANKS1B, is a scaffold protein containing six ankyrin repeats, two sterile alpha motifs (SAMs), and a phosphotyrosine binding (PTB) domain[28–31]. Interestingly, ankyrin repeat- and PTB domain-containing proteins have been implicated developmentally in ependymal differentiation, especially in the neurogenic niche of the lateral ventricle (LV)[32,33]. However, the role of ANKS1 family proteins in regulating the molecular dynamics of the SDAs and the resultant coordinated beating of cilia has not been studied.

In this study, we show a critical role for dynamic changes in FOP localization for inducing BF assembly and in the resultant coordinated beating of motile cilia. Furthermore, we show that the ANKS1A PTB adapter protein is important in the biogenesis and maintenance of polarized SDAs via its interaction with FOP.

## Results

**ANKS1A interacts with FOP in SDAs.** In a previous study, we demonstrated that E1 cells lining the brain ventricles prominently express *ANKS1A* during brain development[34]. Here we investigated the presence of ANKS1A in the lateral wall (LW) of the LV. By performing X-gal staining of LWs from *ANKS1A*[+/lacZ] mice, we found strong *ANKS1A* expression during the first 10 days of postnatal (P) brain development that is gradually reduced after P20 (Fig. 1a) but never entirely lost. *ANKS1A* is still detectable in the LWs of the adult brain (Supplementary Figs. 1a and 5c). Next, we examined the subcellular localization of ANKS1A in primary ependymal cells derived from the LW. We found strong ANKS1A expression near multiple BBs (Supplementary Fig. 1b) that was undetectable in the ependymal cells of *ANKS1A*[lacZ/lacZ] mice (from now, referred to as *ANKS1A* knockout (KO) mice). Using three-dimensional structured illumination microscopy (3DSIM), we confirmed the BB-specific localization of endogenous ANKS1A in the LW of wild-type (WT) mice, and the specificity of the antibody was also validated in the LW of *ANKS1A* KO mice (Fig. 1b). We further used in utero electroporation (IUE) to express a recombinant ANKS1A protein tagged with the N-terminal portion of green fluorescent protein (GFP) (ANKS1A-VN) (Supplementary Fig. 1c). We observed that ANKS1A-VN was prominently detectable in immature cells, which displayed randomly oriented FOP staining as in the images (Supplementary Fig. 1c, middle panels). We then used 3DSIM to map the position of ANKS1A relative to FOP in the immature cells (Fig. 1c, the first panels). This analysis revealed that ANKS1A is localized in a FOP-positive ring structure, reminiscent of SDAs[27,35]. To minimize the anisotropic distortion of randomly oriented BB images, we collected FOP-positive ring images and then used a linear 3DSIM method to assess the localization of ANKS1A and other markers relative to FOP (Supplementary Fig. 1d and Fig. 1c). This molecular mapping revealed that, along the longitudinal axis, ANKS1A is located close to CNTRL- and ODF2-positive regions but distant from CEP164-positive regions (Fig. 1d, f). We also detected ANKS1A-positive puncta in the FOP-stained rim of SDAs rather than in the lumen along the transverse axis (Fig. 1e, f). In addition, our co-immunoprecipitation experiment showed that ANKS1A associates with the FOP-CEP350 complex in the LW lysates from mice at P4 (Fig. 1g). Interestingly, ANKS1A antibodies proportionally immunoprecipitated much more FOP than ANKS1A, suggesting that ANKS1A interacts with a large protein complex containing a much higher stoichiometric ratio of FOP. Consistently, our bimolecular fluorescent complementation

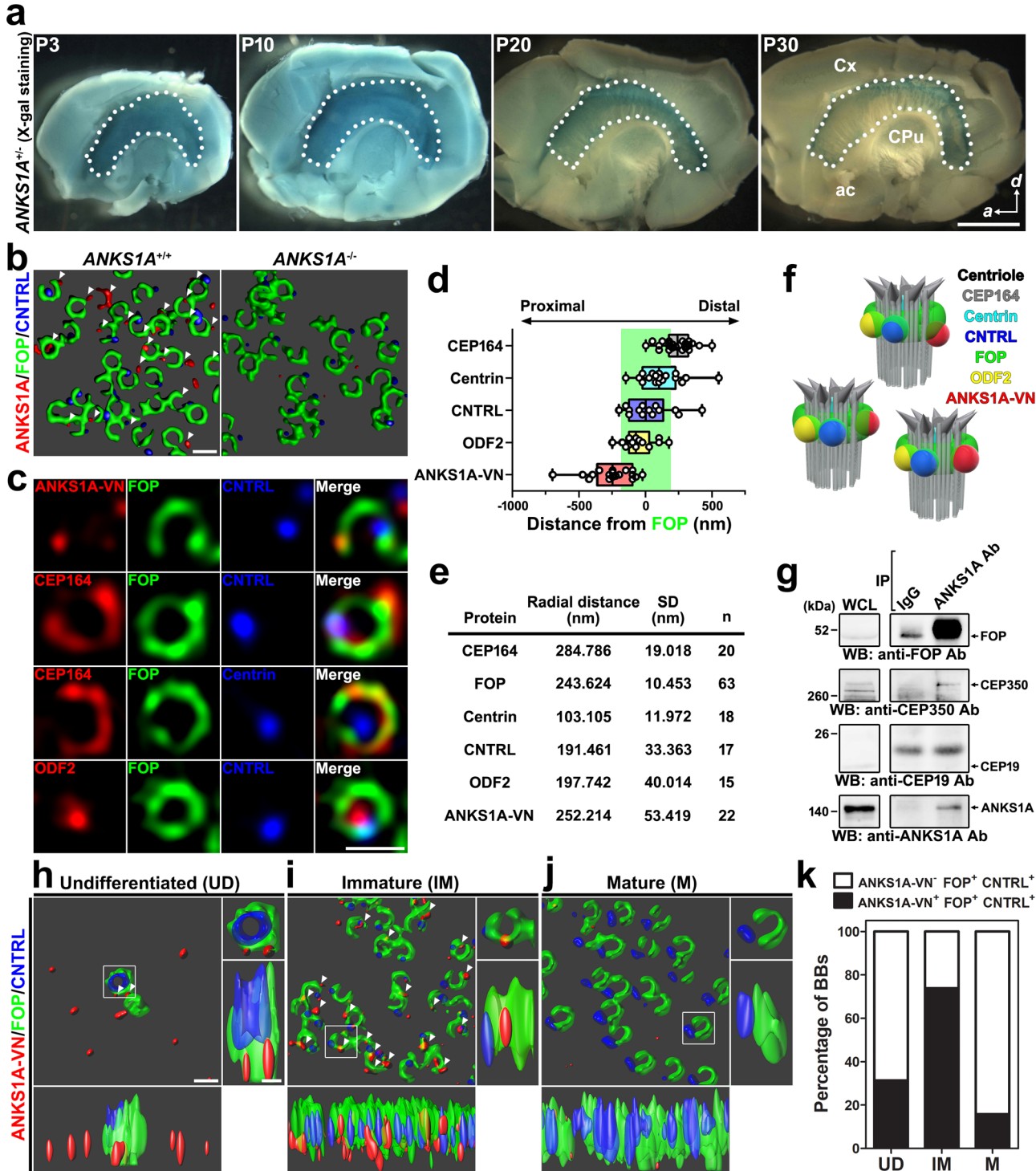

**Fig. 1 Specific localization of ANKS1A to FOP-positive SDAs. a** LWs (marked by dotted lines) with other forebrain tissues were subjected to X-gal staining. Scale bar, 500 μm. Cx cortex, CPu caudate putamen, ac anterior commissure. **b** The LWs were co-stained with ANKS1A, FOP, and CNTRL antibodies. The 3D SIM images were analyzed by the Imaris software. **c** 3DSIM micrographs of representative BBs. *ANKS1A-VN* was injected into the LV at E16.5 and subjected to in utero electroporation. The LWs at P4 were then co-stained for FOP and various BB markers with antibodies. Scale bar for **b**, **c**, 500 nm. **d** A median line and upper and lower quartile are presented in box and whisker plot of the axial distances of BB proteins from immature E1 cells. CEP164, $n = 24$; Centrin, $n = 19$; CNTRL, $n = 14$; ODF2, $n = 15$; ANKS1A-VN, $n = 18$. A green-shaded box marks a FOP-stained region brighter than 30% of the maximum intensity. **e** Table showing the radial distances of BB proteins from immature cells. The center of the FOP-positive ring image was used as a reference for the radial distances of BB proteins; $n$ indicates the number of BBs used for statistical analysis. SD standard deviation. **f** Cartoon depiction of the BB proteins shown in **d**, **e**. **g** Whole-cell lysates (WCL) were prepared from six LWs at P4 and then the protein complexes were precipitated with the C-terminal specific antibody to detect the ANKS1A-associated proteins. **h–j** 3DSIM micrographs of BBs in E1 cells representing three different developmental stages. White arrowheads mark ANKS1A-VN present in FOP-stained SDAs. Scale bar for **h–j**, 500 nm. **k** Data shown in **h–j** were quantified with the total number of the double-positive FOP+CNTRL+ BBs set to 100%. Then the percentage of triple-positive ANKS1A+FOP+CNTRL+ BBs was calculated. Undifferentiated (UD), $n = 67$; immature (IM), $n = 195$; mature (M), $n = 227$ where $n$ indicates the number of BBs analyzed.

analysis revealed that two or more identical FOP proteins associate with each other (Supplementary Fig. 1e). Together, we conclude ANKS1A interacts with areas positive for FOP, a key component of SDAs.

With respect to the various stages of E1 cell development, we found in a 3DSIM analysis that ANKS1A is only partially co-localized with FOP in immature cells and rarely observed in mature cells (Fig. 1h–j). We therefore hypothesized that ANKS1A may play a role in the changes observed in SDAs as they develop. To test this hypothesis, we used FOP and CNTRL staining to delineate the transformation of SDAs in undifferentiated, immature, and mature E1 cells (Fig. 1h–j). We found two FOP-positive centrioles in undifferentiated E1 cells, in which only one centriole had an inner CNTRL-stained ring (Fig. 1h). We also observed immature E1 cells having numerous BBs, in which most of the FOP-stained ring images contained small FOP-negative regions with random orientations (Fig. 1i). In these cell types, we not only found CNTRL-positive puncta rather than ring structures but also ANKS1A-positive puncta in FOP-positive SDAs (Fig. 1k). In an analysis, for example, 74% of the double-positive FOP⁺CNTRL⁺ SDAs were co-localized with ANKS1A puncta, while 60% of ANKS1A puncta among the triple-positive SDAs were in the regions of reduced FOP staining (Supplementary Fig. 1f, g). The mature E1 cells, in contrast, showed apical BBs with extensive FOP-negative regions in alignment with the direction of the CNTRL puncta (Fig. 1j).

**ANKS1A has a role in dynamic changes in SDAs**. We found that, as E1 cells mature, their FOP-negative regions gradually fill with ODF2, a critical scaffold for initiating BF development (Fig. 2a). Here we refer to the FOP-stained region with respect to the BF as the "basal rim" of the SDA (Fig. 2b). A previous study divided the BF assembly of motile cilia into two regions, I and II[20,26]. For immature cells, we observed CNTRL in close proximity to ODF2 in the future BF region I. In mature cells, in contrast, CNTRL and ODF2 were separated from one another and localized to regions II and I, respectively (Fig. 2a, b). We therefore concluded that SDAs undergo a dynamic transition from an unpolarized ring shape to the more polarized architecture observed during E1 cell maturation.

Next, we investigated ANKS1A's role in regulating these dynamic changes in SDAs. We used ANKS1A KO mice to prepare LWs (P40–P45) for 3DSIM analysis. In WT mice samples, the central angle of the FOP-negative region averaged 100° (Fig. 2c, d). In ANKS1A KO mice, however, the basal SDA rim was expanded with the central angle of the FOP-negative region averaging only 70° (Fig. 2c, d). We also found CEP19, which is a known FOP interactor[27,35], localized to the basal rim just like FOP. As we observed with the FOP-negative region, we found ANKS1A KO mice show a similar reduction in the central angle of the CEP19-negative region (Supplementary Fig. 2a, b). Together, these results suggested to us that the underlying BF architecture is affected by ANKS1A. To test this possibility, we used ODF2 and CNTRL antibodies to examine the BF architecture (Fig. 2e). We found similar levels of CNTRL in region II between WT and ANKS1A KO samples but reduced levels of ODF2 in region I in ANKS1A KO samples (Fig. 2e). We further quantified the relative ODF2 level by calculating the ratio of total ODF2 to total CNTRL intensity. This revealed reduced recruitment of ODF2 to the BF architecture in ANKS1A KO mice (Fig. 2f), suggesting a reduced size of region I. Nevertheless, our experiments using CNTRL and CEP164 antibodies showed no significant differences in number, density, or apical docking of BBs per cell between ANKS1A WT and KO samples (Fig. 2g and Supplementary Fig. 2c–f).

To confirm this altered BF architecture in ANKS1A KO samples, we performed a serial section transmission electron microscopy (TEM) comparison (Fig. 2h). Our initial analysis with 50-nm-thick serial sections revealed no significant difference in the relative BF volumes in ANKS1A KO samples compared to ANKS1A WT samples (Supplementary Fig. 2g). We also scrutinized approximately 100 BBs from WT or KO samples (1.5-month-old littermates), indicating that the 9 + 2 ciliary MT structure, distal appendage, and triplet MT structure were not disturbed in KO (Supplementary Fig. 2h). Nevertheless, the indices for the various BF structural features such as height and central angle were significantly different between WT and KO samples (Fig. 2i–k). These results strongly suggest that the BF assembly in ANKS1A KO samples form an unstable cone shape with a narrow base and an elongated height.

**ANKS1A deficiency disrupts the coordinated beating of motile cilia**. The plane-polarized nature of the BF is implicated in the coordinated beating of its motile cilia[3,6,7]. We therefore asked whether the motile cilia in ANKS1A KO samples show defects in growth pattern or function. We performed a whole-mount staining of the well-known cilia markers glutamylated and acetylated tubulin[36]. We compared the ciliary growth pattern in the anterior–dorsal (AD), anterior–ventral (AV), and posterior–medial (PM) regions around the LW adhesion area (Fig. 3a)[6]. At P20, in the AD region, each cell in the WT group formed a well-organized ciliary bundle aligned with the direction of CSF flow (Fig. 3b, c, left panels). In the cilia of ANKS1A KO mice, however, we observed marked disorganization and a lack of proper bending or bundling (right panels). Importantly, scanning electron microscopic (SEM) study also revealed that the growth of multi-cilia was not significantly disturbed in the ANKS1A KO samples (Fig. 3d).

Next, we were able to visualize CSF flow by placing fluorescent beads on live LWs and tracking the pattern of bead movement in the AD region (schematic in Fig. 3a). Typically, we observed beads flowing in the posterior to anterior and dorsal to ventral directions (defined as AD–AV flow; Fig. 3e, f and Movie 1). WT LW showed a typical AD–AV flow pattern at a rate of approximately 225 µm/s (40 beads analyzed from three mice; Fig. 3e, top panels; Fig. 3f, left panel; Fig. 3g). In ANKS1A KO LW, we observed an abnormal and congested bead flow pattern with a 7-fold speed reduction (approximately 30 µm/s; 40 beads analyzed from three mice; $p < 0.0001$; Fig. 3e, bottom panels; Fig. 3f, right panel; Fig. 3g). Together, we conclude that loss of ANKS1A impairs SDA development, interfering with ciliary organization and coordinated ciliary beating.

**ANKS1A deficiency leads to a rotational orientation defect in BBs**. We next asked whether BBs in ANKS1A KO samples show disorganized rotational planar polarity. We used FOP and γ-Tubulin antibodies to stain LWs from P40–P50 samples, allowing us to observe rotational polarity at the cell and tissue levels (Fig. 4a). In the AD region, we drew vectors (black arrows) from FOP-positive areas to the nearest γ-Tubulin-positive areas to indicate the direction of CSF flow (Fig. 4b). The level of γ-Tubulin in BF region II was not different between WT and ANKS1A KO samples. We calculated each BF angle and obtained a mean BF angle within each cell post-normalization of 225° for both WT and KO samples. This hypothetical BF angle is consistent with the direction of CSF flow in the AD region. In the histogram plotting the distribution of the normalized BF angles (Fig. 4c), clustering around the 225° peak indicates normal BF alignment. In WT samples, we found that ninety-two percent of

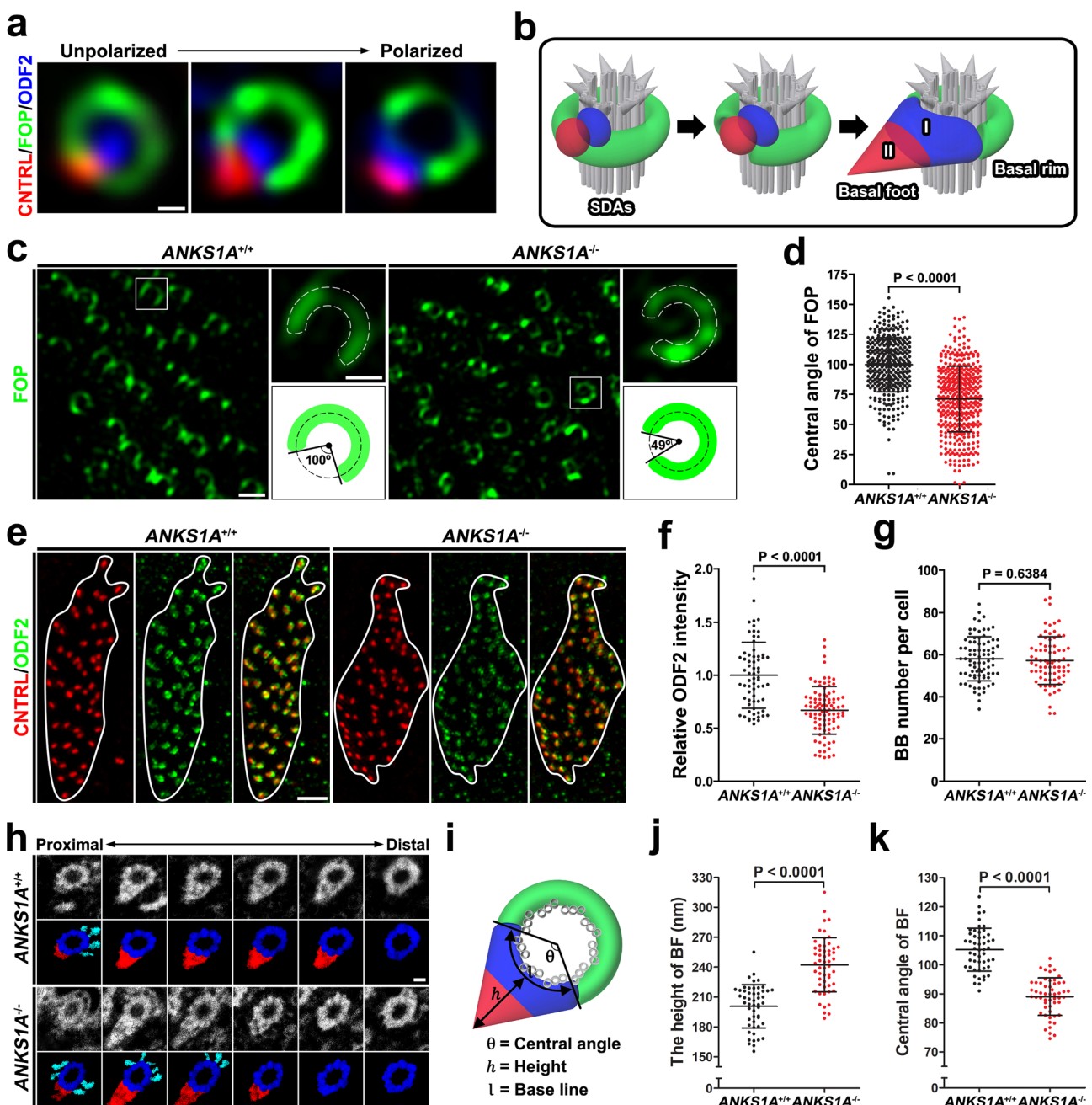

**Fig. 2 ANKS1A loss results in partially impaired SDAs. a** Representative 3DSIM images showing the gradual transition of SDAs from immature to the mature stage. Scale bar, 100 nm. **b** Cartoon depiction elucidating the dynamic change of SDAs from an unpolarized to a polarized state. **c** 3DSIM images were obtained from the anterior–dorsal (AD) region of LWs, and the BB patches were then magnified for further analysis. Schematic beside each panel is for the ×3.5 magnified white box and depicts the central angle of the FOP-negative region. Scale bar, 500 nm. **d** Data in **c** were quantified. Data are depicted as mean ± SD. Each point on the graph represents the central angle of the FOP-negative region. For the *ANKS1A*$^{+/+}$ group, $n = 356$ BBs from 3 mice; for *ANKS1A*$^{-/-}$, $n = 412$ BBs from 3 mice. **e** Experiments were performed as described in **c**, except that CNTRL and ODF antibodies were used. The boundary of each BB patch is outlined in white. Scale bar, 2 μm. **f** Data in **e** were quantified. Data are presented as mean ± SD. Each point on the graph represents the relative level of ODF2 intensity per BB patch. For the *ANKS1A*$^{+/+}$ group, $n = 68$ cells from 2 mice; for *ANKS1A*$^{-/-}$, $n = 96$ cells from 3 mice. **g** Number of BBs per cell was quantified based on their CNTRL staining, as illustrated in Supplementary Fig. 2c. Data represent mean ± SD. For the *ANKS1A*$^{+/+}$ set, $n = 81$ cells from 3 mice; for *ANKS1A*$^{-/-}$, $n = 77$ cells from 3 mice. **h** A ribbon of consecutive sections (50 nm) was placed on a one-hole grid and subsequently analyzed by TEM. Scale bar, 100 nm. **i** Schematic for clarifying the indices that represent the structural morphology of each BF. **j**, **k** The generated micrographs were aligned and used to measure the indices described in **i**. Data represent mean ± SD. For the *ANKS1A*$^{+/+}$ group, $n = 51$ BBs; for *ANKS1A*$^{-/-}$, $n = 54$ BBs.

BF angles clustered within 45° of the mean angle (black line), but this fell to 69% in KO samples (red line) (Fig. 4c). The circular standard deviation (CSD) of these BF angles (error bar on the graph) from the mean angle for each cell was much larger in the KO samples (CSD = 58°; $n = 5635$ BB from 278 cells, 4 mice;

$p < 0.0001$) than in the WT samples (CSD = 27°; $n = 4937$ BB from 234 cells, 4 mice) (Fig. 4c).

To confirm the intracellular rotational polarity defects in *ANKS1A* KO mice, we used TEM to observe the BF protruding laterally from the side of each BB. As expected, the BF positions

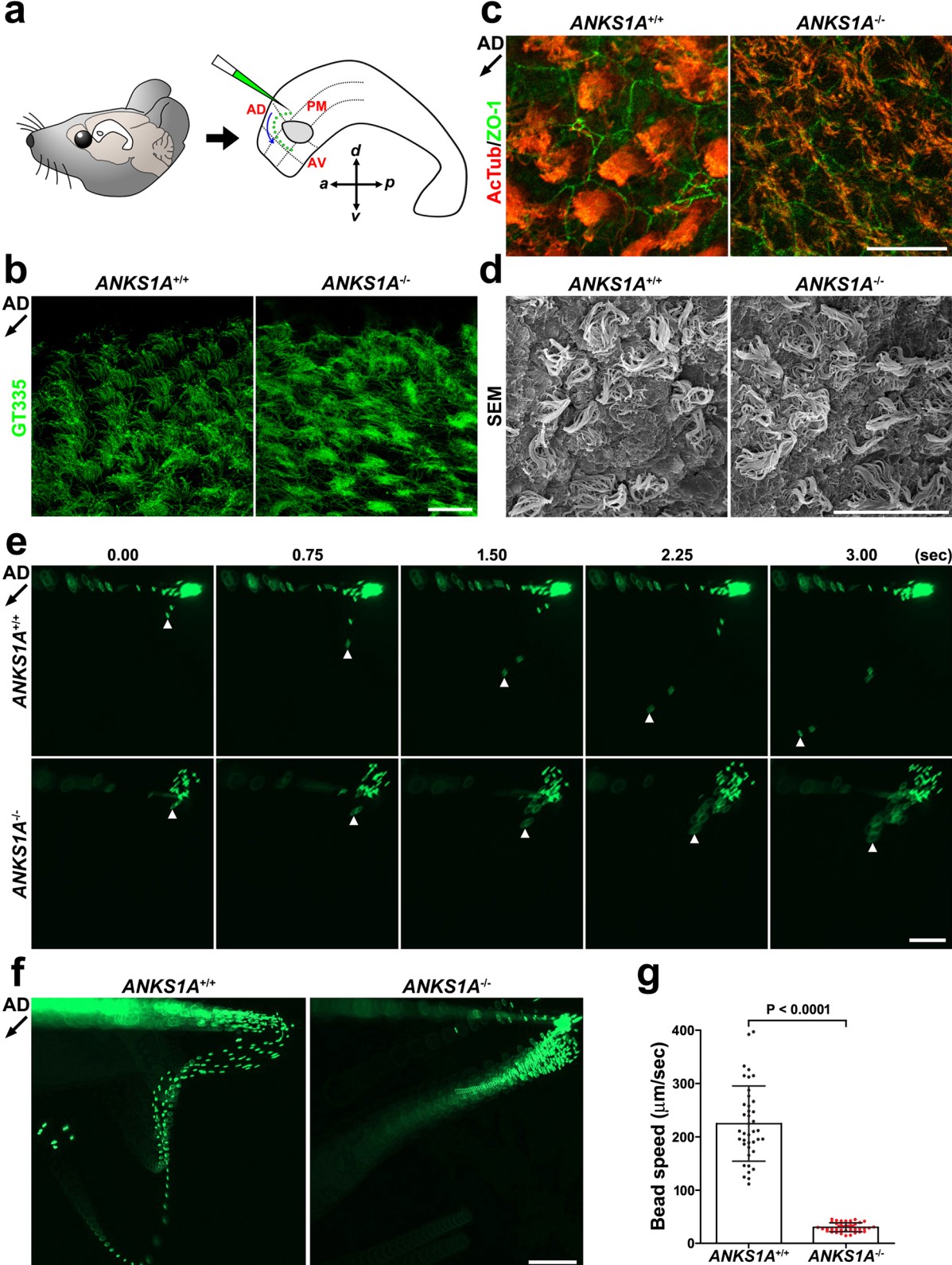

**Fig. 3 ANKS1A loss results in uncoordinated beating of motile cilia. a** Schematic of LWs showing three regions around the adhesion area (marked by a circle) or the movement of fluorescent beads (green) on a live LW. AD anterior–dorsal, AV anterior–ventral, PM posterior–medial. **b, c** Whole-mount staining of LWs using GT335 (glutamylated tubulin) or AcTub (acetylated tubulin) antibodies at P20. The black arrow (~225°) indicates the direction of CSF flow in the AD region. The abnormal phenotype of KO ependymal cilia was observed from at least five different pairs of WT and KO littermates. Scale bar for b, c, 20 µm. **d** SEM analysis of LWs from littermates at P30. Scale bar, 20 µm. This result was reproducibly observed from at least three different pairs of WT and KO littermates. **e** High-speed video imaging analysis of the same fluorescent bead at different time points. **f** Thirty-five consecutive frames of Supplementary Movie 1 were merged into a single picture. Scale bar for **e**, **f**, 200 µm. **g** Data presented in **d** were used to calculate an average bead speed based on at least three independent experiments. Data represent mean ± S.D. Each point on the graph represents the speed of an individual bead.

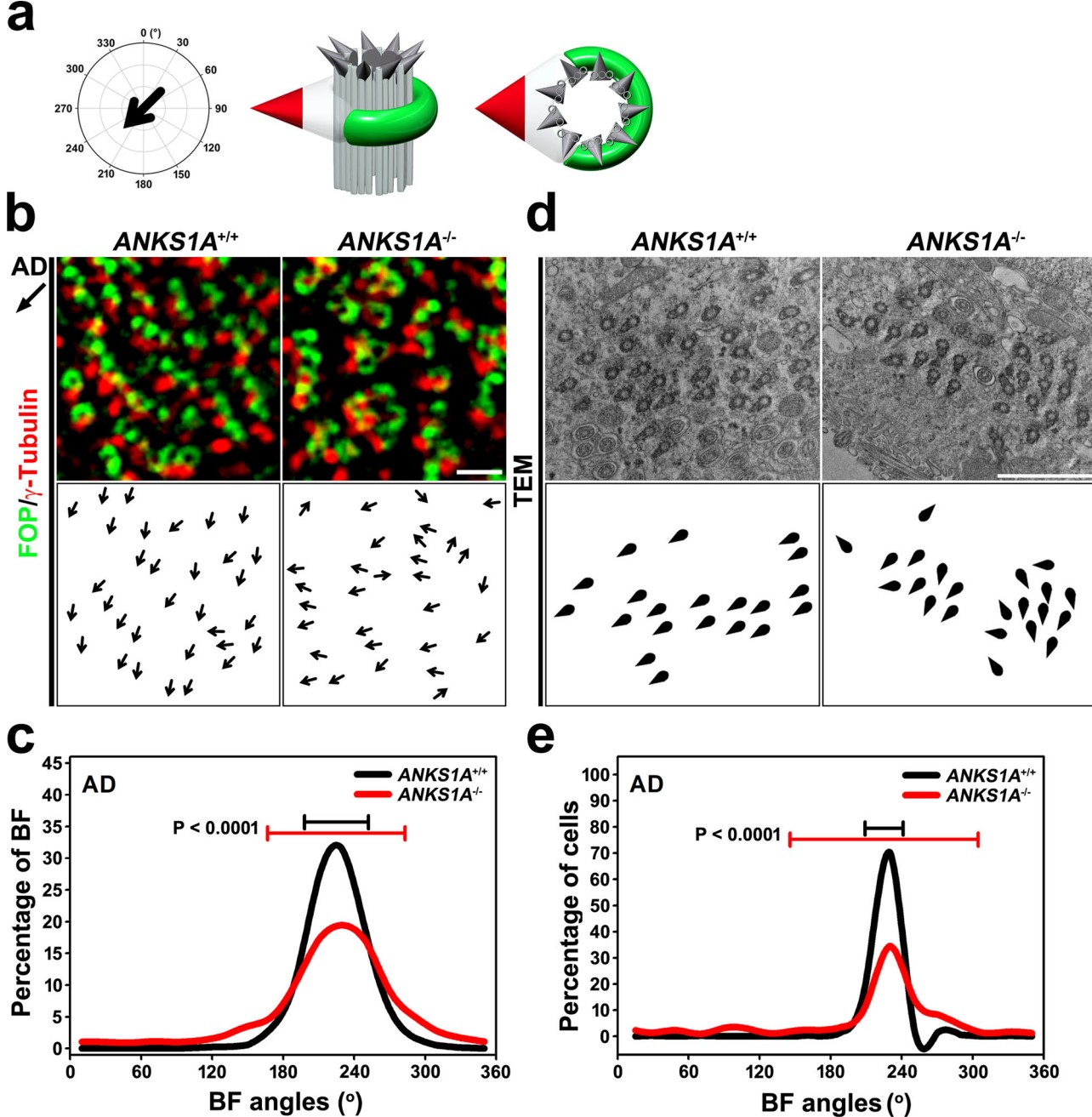

**Fig. 4 ANKS1A loss results in defective rotational polarity. a** Schematic depicting the flow of CSF (225°) in the AD region. Double staining for FOP (green, SDA marker) and γ-Tubulin (red, BF marker) to visualize rotational polarity. **b** Images of the AD region obtained by 3DSIM, and BBs magnified for rotational polarity analysis (top panels). Each black arrow points from the FOP staining toward the γ-Tubulin-positive dot, indicating the rotational polarity axis of each individual BF (bottom panels). Scale bar, 1 μm. **c** Histogram showing the distribution of the rotational vector angles. Error bars on the graph represent CSD. **d** En face electron micrographs from the apical surface of *ANKS1A* WT and KO samples (top panels). Schematics showing the BBs (circular) with BFs (triangular) appear in the bottom panels. Scale bar, 2 μm. **e** The TEM data shown in **d** were used to draw histograms. Error bars on the graphs represent CSD.

were well-aligned with respect to CSF flow in WT samples, but not in *ANKS1A* KO samples (Fig. 4d). Ninety-nine percent of BFs were clustered within 45° of the mean angle in WT samples (black line), but this fell to 71% in KO samples (red line) (Fig. 4e). This TEM analysis also confirmed the increased CSD for the BF angles with respect to the mean angle in KO samples (CSD = 74°; $n$ = 649 BB from 62 cells, 3 mice; $p < 0.0001$) compared to WT samples (CSD = 16°; $n$ = 575 BB from 61 cells, 3 mice).

To analyze tissue-wide rotational polarity, we averaged the rotational angles of each BF for each cell. This mean value was

obtained from approximately 250 cells, normalized to 225°, and then their distribution was plotted on a histogram (Supplementary Fig. 3a, b). This tissue-wide analysis revealed that in 95% of cells, the rotational angle clustered within 45° of the mean angle for both WT and KO samples (Supplementary Fig. 3b). In addition, the CSD of the rotational angle for each cell did not differ significantly between WT (CSD = 35°; 234 cells, 4 mice) and KO samples (CSD = 33°; 278 cells, 4 mice; p = 0.85) (Supplementary Fig. 3b). Together, these data indicate that abnormal SDA architecture disrupts the establishment of intracellular rotational

polarity in *ANKS1A* KO mice. However, we did not find any significant difference in tissue-wide translational polarity in *ANKS1A* KO animals (Supplementary Fig. 3c–f).

There remained the possibility that *ANKS1A* deficiency results in developmental defects for ependymal cells, thus indirectly affecting the BF maturation. To address this, we performed a quantitative reverse transcriptase polymerase chain reaction (RT-PCR) analysis of total RNA extracts of the LWs from pups at various postnatal stages (Supplementary Fig. 4a–c). As a result, we did not find any significant differences in the expression level of marker genes for progenitors (*KI67*), cell fate determination (*GEMC1* and *MCIDAS*), centriole amplification (*DEUP1* and *CEP152*), cilia growth (*FOXJ1*), and planar cell polarity (*VANGL1*, VANGL2, *FZD3, CELSR1-3*) between WT and KO samples. Additionally, asymmetric membrane localization of *FZD3* and *VANGL1* was not impaired in *ANKS1A* KO samples (Supplementary Fig. 4d). These results strongly suggest that *ANKS1A* deficiency does not affect the overall differentiation programming of ependymal MCCs.

**ANKS1A deficiency disrupts normal SDA architecture**. We observed that fully mature E1 cells in P30 or P60 mice also express *ANKS1A*, suggesting that ANKS1A may play a role in dynamic changes in SDAs even after full maturation (Fig. 1a and Supplementary Fig. 1a). To test this possibility, we took advantage of *ANKS1A*-floxed mice, in which exons 9, 10, and 11 of *ANKS1A* are flanked by two loxP sites (Supplementary Fig. 4e). We combined this floxed *ANKS1A* allele, a floxed *ROSA-STOP* allele, and *ANKS1A-CreER* and induced KO of *ANKS1A* with tamoxifen (TM) (*ANKS1A*^f/+; *ROSA*^+/fSTOP; *ANKS1A-CreER* as control and *ANKS1A*^f/f; *ROSA*^+/fSTOP; *ANKS1A-CreER* as *ANKS1A* iKO) (Supplementary Fig. 4e). To delete the floxed alleles, TM was injected at P5 and P6, and then the resulting brains were analyzed at P15 (Supplementary Fig. 4f and Movie 2). The motile cilia of yellow fluorescent protein (YFP)-labeled cells in controls showed coordinated bending and directionality (left panel). The cilia of YFP-labeled cells in the iKO samples, however, showed a scattered rather than bundled pattern (right panel). Importantly, this abnormality was not observed in YFP-negative neighboring cells, suggesting that *ANKS1A* regulates ciliary function in a cell-autonomous manner. To induce a more intense *ANKS1A* gene ablation, TM was injected into young adult mice for 5 days, and then the resulting LW was examined a week later (Fig. 5a). Strikingly, *ANKS1A* iKO cases showed congested bead flow patterns, indicating abnormal AD–AV flow (Fig. 5b, c and Movie 3). While control samples showed approximately 170 μm/s speed (30 beads analyzed from 3 mice), iKO samples showed a 0.7-fold reduction in speed (approximately 100 μm/s speed) (30 beads analyzed from three mice; $p < 0.001$; Fig. 5d).

We next examined ODF2 and CNTRL levels to determine whether *ANKS1A* iKO samples show the same disorganized BB rotational polarity we saw previously (Fig. 5e). Since the *ANKS1A* iKO samples showed reduced levels of ODF2 (Fig. 5h), we had to enhance the signal for the polarity analysis (Fig. 5f, right panel). Consistent with uncoordinated ciliary beating, iKO samples showed altered intracellular rotational alignment of their BF (CSD = 76°, $n = 7700$ BBs from 178 cells from 3 mice; $p < 0.0001$) compared to control (CSD = 26°, $n = 7679$ BBs from 181 cells from 3 mice) (Fig. 5f, g). Since BF CNTRL levels were similar between control and iKO samples, we used the total CNTRL intensity to normalize the ODF2 levels. This allowed us to show that the relative ODF2 level was significantly reduced in the BB patch of iKO samples (Fig. 5h, i). Together, these data indicate that ANKS1A is critically involved in regulating the molecular dynamics of SDAs even in fully mature cells. These results also

rule out the possibility that ANKS1A deficiency affects normal differentiation programming of ependymal cells with the BF phenotype being a secondary outcome.

**ANKS1A deficiency aggravates brain aging-induced SDA degeneration**. We found that young adult KO mice show only mild SDA structural impairment restricted mainly to BF region I. We postulate that this partial SDA defect is insufficient to affect overall BB stability in KO mice (Fig. 2g and Supplementary Figs. 2d–f and 4g). It remains possible, however, that aged mice show SDA defects that become more unstable and more damaged with time. Such additive effects would exert increasingly damaging influence on BB stability and on motile cilia in general from the constant shear stress. To test this possibility, we analyzed *ANKS1A* KO mice from 18 to 22 months of age. The lifespan of *ANKS1A* KO mice did not differ from their WT littermates, but *ANKS1A* KO did show clear evidence of histological degeneration in the LWs. Not only were motile cilia less abundant (Fig. 6a, top panels; Fig. 6b), but even the number of BBs was reduced (Fig. 6a, middle panels; Fig. 6c). In addition, the actin network for each BB patch was significantly less dense in both the apical and subapical regions (Fig. 6a, bottom panels; Fig. 6d, e). We reasoned that this deterioration of E1 cells may stem from more severe structural changes in their SDAs (Fig. 6f). Interestingly, the average central angle of the FOP-negative region in WT samples was 86° (Fig. 6h), significantly less than the 100–110° of the young adult samples (Fig. 2d and Supplementary Fig. 2b). The central angle of the FOP-negative region in *ANKS1A* KO was even further reduced to an average of 47° (Fig. 6h). In *ANKS1A* KO samples, some of the FOP-stained basal rims showed both fragmentation and reduced FOP signal (Fig. 6g, i). Importantly, ODF2 and CNTRL showed a marked change in distribution from a polarized BF protein to an unpolarized one intercalated into the SDA ring structure. To confirm whether the size of BFs is reduced in *ANKS1A* KO cases, we performed a serial section TEM structural analysis. Analysis with 50-nm-thick serial sections revealed that the relative BF volume of *ANKS1A* KO samples is significantly reduced compared with that of *ANKS1A* WT samples (Fig. 6j, k). Together, our data indicate that ANKS1A in concert with FOP protects SDAs from structural degeneration during brain aging. We also performed magnetic resonance microimaging to examine brain ventricle size in aged mice (Supplementary Fig. 5a and Movie 4). This analysis showed that ~50% of the *ANKS1A* KO brains have enlarged LVs (Supplementary Fig. 5b). Although this is a significant portion of the KO samples, for unknown reasons some samples showed no change in ventricle size. Nevertheless, we observed 100% penetrance for the histological phenotypes, such as loss of motile cilia or BBs in aged mice.

To further examine whether ANKS1A has a role in the stability of adult BBs, we first confirmed that LWs from *ANKS1A*^+/lacZ mice at 3 months of age do express *ANKS1A* (Supplementary Fig. 5c). We then generated iKO mice by injecting TM into 3-month-old mice and then analyzed their brains at 20–22 months (Supplementary Fig. 5d). As shown in Supplementary Fig. 5e–g, both BBs and motile cilia were significantly reduced in aged iKO mice, strongly supporting a role for ANKS1A in stability of adult BBs.

We further examined whether the MT network was disturbed as a consequence of ANKS1A loss (Fig. 7). Neither α-Tubulin staining nor serial TEM analysis showed any significant differences for the MT networks between young adult WT and KO mice (Fig. 7a, b, e–h). However, aged KO mice had a significant reduction in the extent of MT networks in regions where BBs were defective (Fig. 7c, d, i–k and Movie 5). Unlike the MT networks and for unknown reasons, the intermediate

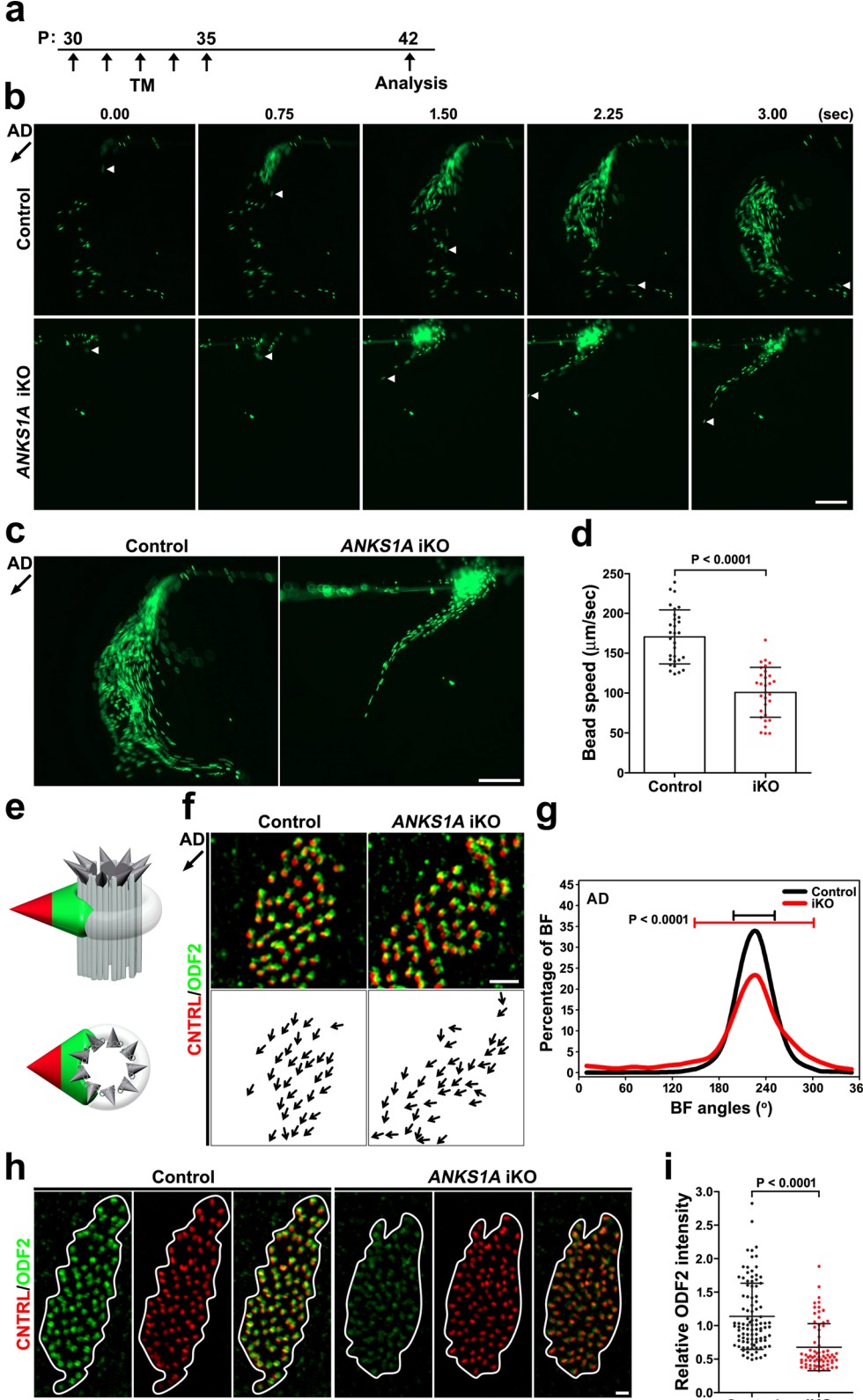

**Fig. 5 Inducible ANKS1A ablation results in a rotational polarity defect, affecting the SDAs. a** Control and iKO mice were generated via five separate daily TM injections from P30 to P35. **b** The mice were sacrificed at P42 for live LW preparation, and the experiments were performed essentially as described in Fig. 3. **c** Thirty-five consecutive frames of Supplementary Movie 3 were merged into a single picture. Scale bar for **b**, **c**, 200 μm. **d** Data presented in **b** were used to calculate an average bead speed based on three independent experiments. Data represent mean ± SD. **e** Double staining for both ODF2 (green) and CNTRL (red) permitted the visualization of rotational polarity. **f**, **g** Experiments were performed as described in Fig. 4b, c. **h**, **i** Experiments were performed as described in Fig. 2e, f. Scale bar for **f**, **h**, 1 μm. Control set, $n = 99$ cells; *ANKS1A* iKO, $n = 71$ cells.

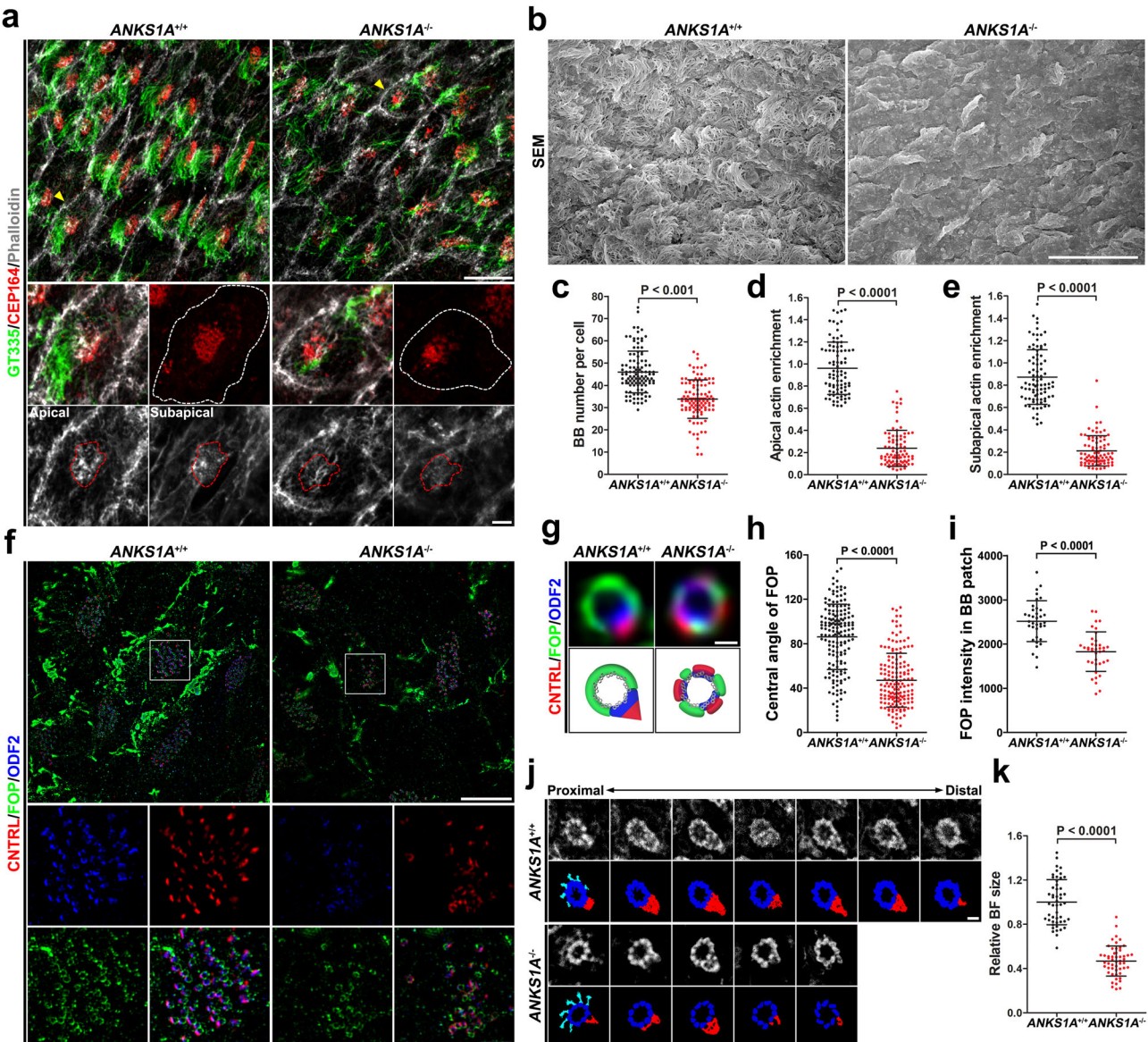

**Fig. 6 ANKS1A loss results in severe SDA degeneration in aged mice. a** Confocal microscopic analysis of LWs from littermates at 22 months of age (top panels). A representative cell (marked by yellow arrowheads) in each panel is enlarged to reveal motile cilia (green), BBs (red) (middle panels), and F-actin (gray) at the apical and subapical levels (bottom panels). The cell border and actin network around a BB patch are outlined with white and red dotted lines, respectively. **b** An SEM analysis of LWs from littermates at 22 months of age. Scale bar for **a**, **b**, 10 µm. **c–e** Data from **a** were quantified. The mice used for this quantification ranged from 18 to 22 months of age. For **c**: *ANKS1A*$^{+/+}$ set, $n = 110$ cells from 4 mice; *ANKS1A*$^{-/-}$, $n = 113$ cells from 4 mice. For **d**, **e**: *ANKS1A*$^{+/+}$ group, $n = 87$ cells from 4 mice; *ANKS1A*$^{-/-}$, $n = 85$ cells from 4 mice. **f** 3DSIM micrographs of LWs depict triple staining with FOP, ODF2, and CNTRL. White boxes indicate the regions of ×3 magnification that reveal SDA staining with each antibody (second and third panels). Scale bar, 10 µm. **g** Representative SDA images (top) together with their cartoon depiction (bottom). Scale bar, 200 nm. **h** The central angles were calculated. *ANKS1A*$^{+/+}$ set, $n = 164$ BBs from 3 mice; *ANKS1A*$^{-/-}$, $n = 160$ BBs from 3 mice. **i** Data in **f** were quantified. Data represent mean ± SD. *ANKS1A*$^{+/+}$ group, $n = 50$ cells; *ANKS1A*$^{-/-}$, $n = 56$ cells. **j**, **k** A ribbon of consecutive sections (50 nm) was placed on a one-hole grid and subsequently analyzed by TEM. The resulting micrographs were aligned to measure the overall BF size. Data represent mean ± SD. *ANKS1A*$^{+/+}$ set, $n = 50$ BBs; *ANKS1A*$^{-/-}$, $n = 54$ BBs. Scale bar, 100 nm.

filaments were not significantly disturbed even in the aged KO mice (Fig. 7i, j).

**ANKS1A's role in SDA polarization is highly conserved in vertebrate evolution**. To determine whether ANKS1A's role with respect to SDAs has been conserved in vertebrates, we examined the *Xenopus laevis* genome. We found that while the *Xenopus* ortholog shows 58% identity with full-length mouse ANKS1A, full-length mouse ANKS1B is absent in *Xenopus* (Supplementary

Fig. 7a–d). We next chose the epidermis of *Xenopus* as another excellent system to study the development of MCCs[1]. We found the FOP antibody that was so effective at staining the mouse LW can also be used to visualize the basal rim of SDAs in the epidermis of *Xenopus* (Fig. 8a, c). For *anks1a* ablation, we designed a morpholino (MO) oligonucleotide to repress the translation of *Xenopus anks1a* mRNA (Supplementary Fig. 7e). Compared with control samples, microinjection of *anks1a*-MO into *Xenopus* embryos produced partial defects in the number and density of BBs at both stage 24 and stage 35 of embryonic development

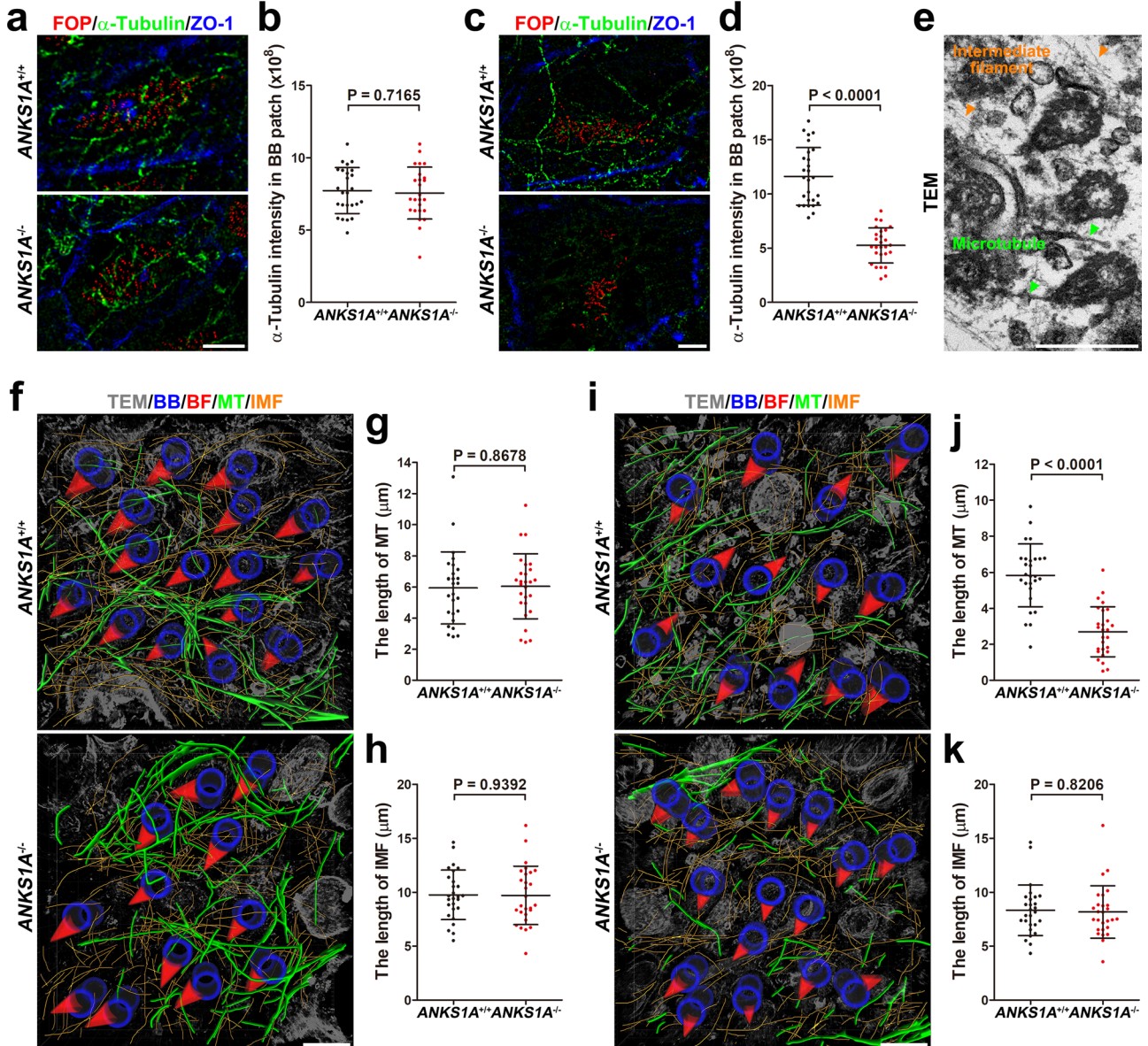

**Fig. 7 ANKS1A deficiency results in a marked loss of the microtubule network in aged mice. a–d** 3DSIM images of the MT networks stained with alpha-tubulin antibody for both young (P45) and aged (20-month old) mice. **a, b** *ANKS1A*⁺/⁺ set, $n = 6$ cells from 2 mice; *ANKS1A*⁻/⁻, $n = 5$ cells from 2 mice. **c, d** *ANKS1A*⁺/⁺ set, $n = 9$ cells from 3 mice; *ANKS1A*⁻/⁻, $n = 9$ cells from 3 mice. Scale bar for **a**, **c**, 5 μm. **e** A representative TEM image illustrating the intermediate filament and microtubule. Scale bar, 500 nm. **f–j** 3D reconstruction of the microtubule and intermediate lattice. The image of the microtubules (green) and intermediate filament (orange) was reconstructed and superimposed on a single plane. For **g, h** (P45 mice), *ANKS1A*⁺/⁺ set, $n = 10$ cells from 2 mice; *ANKS1A*⁻/⁻, $n = 10$ cells from 2 mice. For **j, k** (20-month-old mice), *ANKS1A*⁺/⁺ set, $n = 9$ cells from 2 mice; *ANKS1A*⁻/⁻, $n = 9$ cells from 2 mice. Scale bar for **f**, **i**, 500 nm.

(Supplementary Fig. 7f–i). In control samples, at stage 24, the average central angle of the Fop-negative region was 60°, but this was increased to an average of 84° by stage 35 (Fig. 8b, d). In contrast, in the *anks1a* MO-injected embryos, the average central angle was 52° at stage 24, and this was marginally increased to 57° by stage 35 (Fig. 8b, d). Importantly, the direction of the Fop-negative regions remained consistent with the direction of fluid flow in control embryos. In contrast, the same orientations were more disorderly in MO-injected embryos (Fig. 8e). Finally, we found that the ectopic expression of mouse *ANKS1A* is able to rescue partially the MO-induced defects in the *Xenopus* embryos at stage 35 (Fig. 8c–e and Supplementary Fig. 7h, i). These data strongly support our hypothesis that ANKS1A has a role in polarizing SDAs in MCCs across vertebrate evolution (Fig. 8f).

## Discussion

In this study, we have shown that ANKS1A interacts with FOP to regulate dynamic changes in SDAs. In differentiating cells, SDAs undergo substantial changes from unpolarized circles to polarized, raindrop-like structures. The ultimate outcome of *ANKS1A* deficiency is a shrinkage of BF region I and subsequent elongation of region II. Despite the instability of the resulting BF architecture, the BBs are still competent to form motile cilia. It is evident, however, that the resulting partially impaired SDAs have defective rotational polarity that leads to uncoordinated ciliary beating. It is also noteworthy that BF region I shrinks after *ANKS1A* gene ablation. Furthermore, normal SDAs undergo partial structural degeneration during brain aging with BF region I shrinking and the number of BBs decreasing slightly (compare

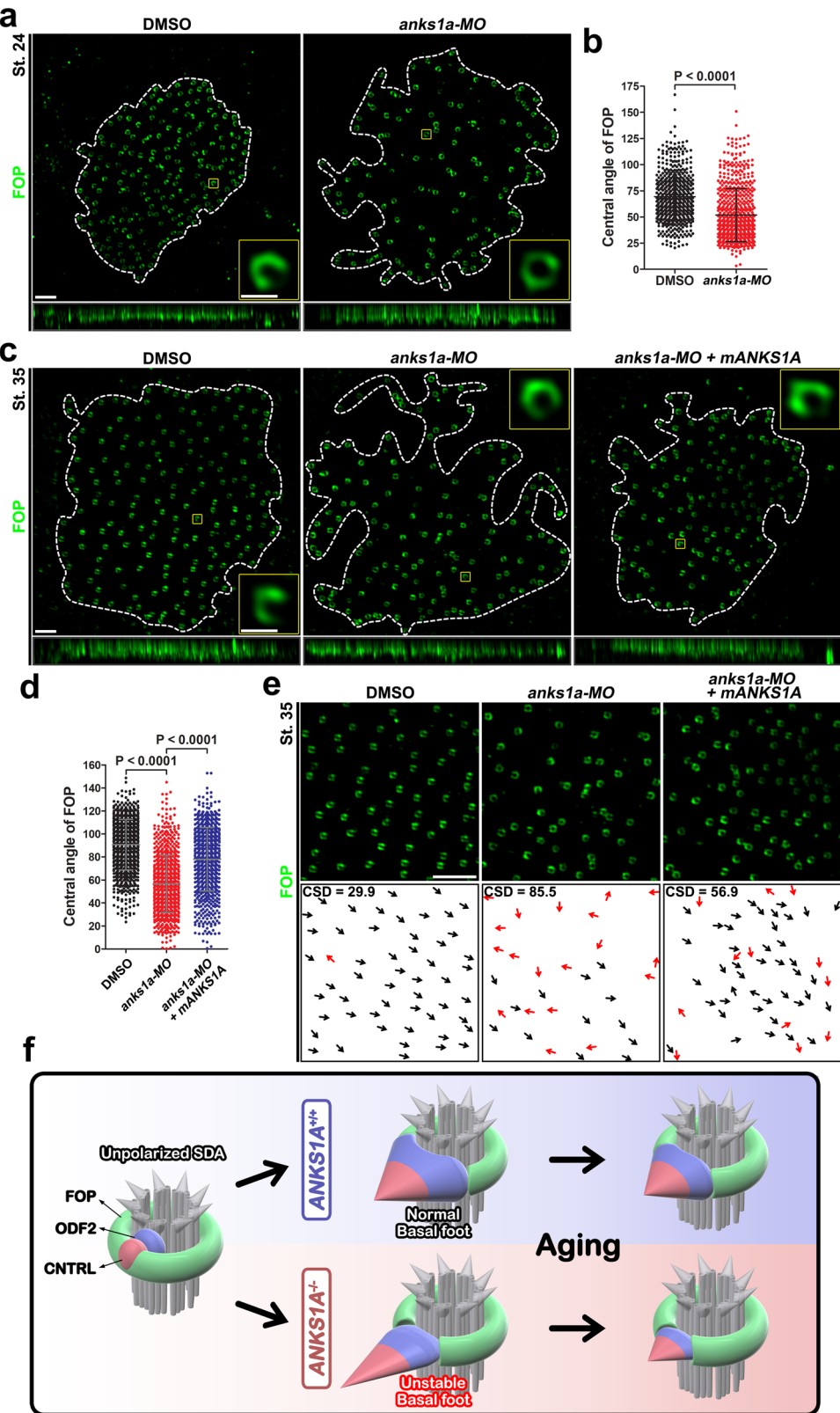

Fig. 2g with Fig. 6c). Importantly, *ANKS1A* deficiency aggravates the natural degeneration of SDAs in aged mice leading to a disintegration of the FOP-positive basal rim and the intercalation of BF proteins into the resulting unpolarized ring structure. The deteriorating SDAs are likely to have BBs surrounded by a more unstable and fragile cytoskeletal network, ultimately leading to detachment of BBs and motile cilia through the long-term exposure to shear stress. Anks1A is also required for BB polarity in *Xenopus*. Together, these data suggest ANKS1A regulates FOP to transform SDAs from an unpolarized state to a more polarized structure essential for rotational polarity.

Our results show that mature SDAs have a unique FOP-negative region that undergoes dynamic changes as it becomes BF region I. This region is absent in undifferentiated cells, appearing

**Fig. 8 ANKS1A function is highly conserved in *Xenopus*. a** 3DSIM images were obtained from the epidermis of *Xenopus* embryos at stage 24. The BB patch outlined with white dotted lines was magnified for further analysis. A representative Fop image at ×7 magnification of the region indicated by the yellow box. The central angle of the Fop-negative region is clearly visible. **b** The data in **a** were quantified. The data represent mean ± SD. Each point on the graph represents the central angle of a Fop-negative region. DMSO injection, n = 501 BBs from 3 embryos; *anks1a*-MO injection, n = 645 BBs from 3 embryos. **c, d** Experiments were performed essentially as described in **a, b**, except that the 3DSIM images were obtained from the epidermis of *Xenopus* embryos at stage 35. DMSO injection, n = 905 BBs from 5 embryos; *anks1a*-MO injection, n = 854 BBs from 5 embryos. *anks1a*-MO plus mouse *ANKS1A-VN* RNA injection, n = 726 BBs from 3 embryos. **e** Experiments were performed as described in Fig. 4b. Scale bar for **a, c, e**, 2 μm. **f** Model in which ANKS1A plays a pivotal role in the polarization of SDAs during the development of MCCs. In the absence of ANKS1A, the SDA BF adopts an unstable architecture and undergoes a gradual degeneration over the course of aging, causing the loss of the BBs and motile cilia.

first in a small form in immature cells. As E1 cells mature, the region gradually grows in size. This FOP-negative region adopts an average central angle of 100–110° in mature cells. The BF is known to associate with the 4, 5, and 6 triplets of a BB, spanning 120° from the center of the BB[37,38]. This is where ODF2 accumulates, filling out the FOP-negative region to form BF region I. ANKS1A deficiency also reduces the size of the FOP-negative region so that ODF2 levels are also significantly reduced. Based on these findings, we hypothesize that ANKS1A is critically involved in the formation of the FOP-negative region and that this region functions as a platform to initiate BF biogenesis. Our results show, however, that *ANKS1A* deficiency does not entirely abolish ODF2 signal in region I. Interestingly, we did not see any significant changes in the levels of CNTRL or γ-Tubulin in region II upon *ANKS1A* deficiency, suggesting that the reduced level of ODF2 still manages to recruit them. The resulting architecture of the BF is reminiscent of an unstable pyramid with a smaller supporting base and a steeper inclination to the apex (Fig. 8f). It remains unclear how ANKS1A mediates the formation of BF region I in concert with FOP. FOP is initially localized to the entire unpolarized SDA, only being restricted to the basal rim as the BF begins to appear. As ANKS1A is frequently localized in transient puncta in small FOP-negative regions (Fig. 1k and Supplementary Fig. 1f, g), we postulate that ANKS1A plays a role in carving the future region I from circular SDAs. It is noteworthy that ANKS1A has multiple protein interaction domains, including ANK repeats, SAM, and PTB, which is typical of a scaffolding protein. In future studies, we plan to evaluate whether ANKS1A recruits proteases that remove FOP to allow for the establishment of BF region I.

In this study, we define the FOP-positive region of SDAs as the basal rim and elucidate its potential function. We confirmed that CEP 19, which interacts with FOP[27,35], is also present in the basal rim (Supplementary Fig. 2a). The basal rim has other notable characteristics. First, the basal rim shows dynamic shape changes depending on cell maturation status. It is a circular structure in undifferentiated cells, which then retracts in immature cells. As maturation proceeds, the basal rim is transformed into a crescent shape. Second, there is an inverse correlation between basal rim size and the polarized structure of the SDAs. As the basal rim expands, region I is reduced, and vice versa. Third, the basal rim remains a stable structure throughout life. The basal rim actually expands significantly during brain aging and does not disappear unless the BF assembly is demolished. These results support our hypothesis that the basal rim provides a strong structural support for the BF architecture. It is likely that the basal rim is tightly linked with BF region I in forming a large supramolecular complex. We hypothesize that it is this structural support that allows the BF architecture to organize the cytoskeletal network and coordinate ciliary beating. Our study suggests BF region I is especially vulnerable to damage. Reductions in BF region I size are accompanied by an expansion of the basal rim, implying that the basal rim is structurally altered to hold more tightly when the BF structure is

less stable. We postulate that this dynamic change in the basal rim is a reversible process as long as ANKS1A is present. In our model, ANKS1A acts as an antagonist of FOP in the basal rim. In the absence of *ANKS1A*, this reversible process does not function properly and the basal rim continues to expand, encroaching on BF region I. Over time, the BF is demolished and FOP is replaced with BF proteins in the basal rim. Therefore, we postulate that ANKS1A interacts with FOP to maintain the proper size of the basal rim and that this process is crucial for supporting and maintaining the BF architecture. Interestingly, our bimolecular fluorescent analysis reveals that two or more identical FOP proteins associate with each other, possibly due to the LisH domain, known to be critical in dimerization and centrosomal localization of FOP (Supplementary Fig. 1e)[39,40]. A GFP-labeled FOP complex recapitulating the basal rim of SDAs would be helpful in investigating the dynamics of SDAs in ependymal cells.

The evidence we have collected indicates that a proper BF assembly is essential for organizing the densely interwoven cytoskeletal network within the BB patch[18,41]. In addition, forceful beating of the cilia induces a stronger cytoskeletal network, suggesting that the BF, cytoskeletal network, and coordinated ciliary beating form a positive feedback loop that establishes the long-term stability of the BB patch. In this respect, *ANKS1A* KO mice offer an ideal model system for investigating the role of polarized SDAs in establishing a positive feedback loop that regulates the stability of the BB patch. Our results show that neither the number of BBs nor the cytoskeletal networks are significantly altered in young adult mice <2 months of age (Figs. 2g and 7a, f). These results strongly suggest that even the affected BF assembly manages to form a positive feedback loop with the stable BB patch. We cannot rule out the possibility, however, that changes in the cytoskeletal network of the BB patch may be too subtle to detect with our experimental protocols. For mice aged >18 months, *ANKS1A* deficiency does have a significant effect on the detachment of motile cilia and their BBs from the apical surface. Actin and MT networks in these BB patches are also markedly disrupted. The most noticeable change is a severe degeneration of SDAs in samples such that BF proteins form an unpolarized ring rather than a polarized structure. This finding suggests that the positive feedback system is markedly disrupted over time and that BBs and motile cilia eventually succumb to shear stress due to their loose and fragile cytoskeletal network. We therefore hypothesize that ANKS1A is crucial for maintaining the polarized structure of SDAs and that this polarization protects the BB patch from degeneration in brain ventricles. Nevertheless, the resulting hydrocephalus is not as severe in older mice as we expected, suggesting that bulk CSF circulation is driven by its production and reabsorption without any stenosis. Additional analysis of CSF contents and adult neurogenesis will help address whether brain homeostasis is also defective in these aged mice.

Several issues must still be addressed concerning the identity and function of our hypothetical SDAs consisting of a basal rim and BF. These include identifying the specific molecular

components of the basal rim and BF region I as well as the kind of proteases that are required for carving out the specific portion of the FOP-stained ring structure where the BF is established. It will also be important to determine whether ANKS1B has a functional redundancy with respect to ANKS1A in affecting ependymal differentiation. Unlike *ANKS1A* KO mice, our results show that knock-down of *anks1a* in *Xenopus* results in more severe BB defects such as a more reduced density of BBs in *Xenopus* epidermis. A potential explanation may be due to the presence of full-length ANKS1B in mice and not in *Xenopus*. Our data does indeed show that *ANKS1B* is expressed in ependymal MCCs in mice (Supplementary Fig. 7j), raising the possibility that ANKS1B provides a functional redundancy to ANKS1A developmentally for maturation of BBs. As such, study of a double KO mice lacking both *ANKS1A* and *ANKS1B* would be insightful in addressing their potential developmental roles for BB maturation. Finally, it will also be helpful to determine whether the role of ANKS1A in E1 cells is also evolutionarily conserved in other MCCs, such as airway tracheal or oviduct cells.

## Methods

**Mice**. For generating *ANKS1A*[+/lacZ] gene trap mice[30,31], embryonic stem (ES) cell clone (CF0537) was purchased from the Mutant Mouse Regional Center, microinjected into C57BL/6 blastocysts, and chimeric males were mated with 129/SvJ females. For generating *ANKS1A*[+/f] mice, ES cell clone (EPD0578_4_D01 containing Anks1[tm1a(KOMP)Wtsi] allele) was purchased from the Mutant Mouse Regional Center, microinjected into C57BL/6 blastocysts, and chimeric males were mated with 129/SvJ females. F1 agouti pups were analyzed for the presence of the transgene, and then FRT-flanked *Neo* cassette was removed by crossing with *FLP* transgenic mice. For generating *Anks1a-CreER* BAC transgenic mice, we used bacterial homologous recombination to insert the PCR-amplified *CreER* DNA into RP24-258K7 BAC[34]. *R26-stop-EYFP* mice were purchased from The Jackson Laboratory. Genotyping was performed using PCR analysis of tail genomic DNA with the following primers: *ANKS1A*[+/lacZ] mice were identified with 5'-TGAAGG CACATGACCCTGAG-3' (forward 1), 5'-ATGTCATAGCTGTTTCCTGT-3' (forward 2), and 5'-ACAGCGTTTGCATCTTGCTG-3' (reverse); Anks1[+/f] mice were identified with 5'-GATTCCTGGAAGCCCGAGAAGAACT-3' (forward) and 5'-TTGAACTCACAGAGCTGTCTGCCTG-3' (reverse); Anks1a-CreER mice were identified with 5'-TGGATAGTGAAACAGGGGCAATGGT-3' (forward) and 5'-TCTCCACCATGCCCTCTACACATTT-3' (reverse); R26-stop-EYFP mice were identified with 5'-GGGAGGGGAGTGTTGCAATACCTTT-3' (forward) and 5'-AAGACCGCGAAGAGTTTGTC-3' (reverse). To produce *ANKS1A* iKO mice, *ANKS1A*[f/+] or *ANKS1A*[f/f] mice were injected intraperitoneally with 4-hydroxytamoxifen (Sigma-Aldrich) in corn oil (75 mg/kg body weight, 10 mg/ml stock solution). All experiments relevant for mice were approved by and were in compliance with the Sookmyung Women's University Institutional Animal Care and Use Committee (SWU-IACUC), Animal and Plant Quarantine Agency, Seoul, Korea. All mice were maintained at the animal facility of Sookmyung Women's University under the strict guideline of the laboratory animal resource department of National Institute of Food and Drug Safety Evaluation in South Korea. The temperature of mouse facility was between 18 and 23 °C and the humidity was between 40 and 60%. Mice were housed in plastic cages on a 12-h light cycle with ad libitum access to water and a standard laboratory diet.

**Xenopus**. *X. laevis* were obtained from the Korean *Xenopus* Resource Center for Research. *Xenopus* embryos were injected after in vitro fertilization of oocytes that was induced by injection of 500 units of human chorionic gonadotropin (Sigma-Aldrich, USA). MO was injected into the animal pole at the 1- or 2-cell stage embryos and cultured in 30% Marc's Modified Ringer's solution. The animal caps (ACs) were then dissected from injected and un-injected embryos at stage 8.0–8.5 and incubated in 1× L-15 growth medium (Gibco/Thermo Fisher) until stage 24 or stage 35 in preparation for RT-PCR. *Xenopus* study was conducted in accordance with the regulations of the IACUC of Hallym University (Hallym2019-81). All the research members attended both the educational and training courses for the appropriate care and use of *Xenopus* at our institutions in order to receive an animal use permit. Adult *X. laevis* were grown in approved containers by authorized personnel for laboratory animal maintenance, with 12-h light/dark cycles at 18 °C according to the guidelines of the Institutes of Laboratory Animal Resources. *xanks1a* MO (Genetools) was the anti-sense oligo deoxynucleotide used for the loss-of-function study. The sequence of *xanks1a* MO is as follows: 5'-AATAACTCCTGTTCTTTCCCCATCC-3'. MO was warmed at 55 °C for 5 min and kept at 37 °C until injection into the embryos. *xanks1a* MO was injected with 150 ng per embryos. Total RNA was isolated from AC explants using RNA-Bee reagent (Tel-Test), and it was treated with DNase I to remove genomic DNA contamination. RT-PCR was performed with Superscript II (Invitrogen) as

described by the manufacturer and with 2 μg total RNA per reaction. PCR was performed according to the following conditions: 30 s at 94 °C, 30 s at each annealing temperature, 30 s at 72 °C, and 20–30 cycles of amplification. The PCR amplification primers for *anks1a* and *odc1* genes are described in Supplementary Table.

**LW dissection and immunohistochemistry**. LWs were isolated from mouse cerebral hemispheres[42,43] and fixed with 4% paraformaldehyde (PFA) and 0.1% Triton X-100 in phosphate-buffered saline (PBS), washed with 0.1% Triton X-100 in PBS three times for 10 min at room temperature. The LWs were incubated with 3% bovine serum albumin (BSA) in PBS for 1 h and treated with primary antibodies diluted in 3% BSA in PBS at 4 °C. The following antibodies were used: rabbit IgG anti-Anks1a (1:100, Bethyl, A303-049A), mouse IgG1 anti-CNTRL (1:100, Santa Cruz, sc-365521), mouse IgG2b anti-FOP (1:250, Abnova, H00011116-M01), rabbit IgG anti-CEP164 (1:200, Atlas Antibodies, HPA037606), mouse IgG2a anti-centrin (1:100, Millipore, 04-1624), rabbit IgG anti-ODF2 (1:100, Atlas Antibodies, HPA001874), mouse IgG1 anti-GT335 (1:200, Adipogen, AG-20B-0020), mouse IgG2b anti-Ac-tubulin (1:200, Sigma Aldrich, T7451), mouse IgG1 anti-γ-Tubulin (1:100, Abcam, ab11316), phalloidin-Atto647N anti-F-actin (1:50, Sigma Aldrich, 65906), mouse IgG1 anti-ZO-1-488 (1:100, Invitrogen, 339188), chick anti-GFP (1:250, Abcam, ab13970), rabbit anti-CEP19 (1:100, Proteintech, 26036-1-AP), mouse IgG2a anti-α-tubulin (1:200, Santa Cruz, SC5286), mouse IgG2a anti-Frizzled3 (1:100, Sigma Aldrich, WH0007976M9), and rabbit anti-Vangl1 (1:100, Atlas Antibodies, HPA025235). After an overnight incubation, the LWs were washed, treated with species-specific secondary antibodies for 1 h, washed with PBS, and mounted in Prolong gold mounting medium (Invitrogen). For IUE, a pregnant female was anesthetized and her uterine horns were exposed. ANKS1A-VN and TdT expression vectors (2.5 mg/ml for each) were injected into the left and right LVs of each embryo, respectively. For electroporation, 5 pulses separated by 950 ms were applied at 45 V for each embryo using the BTX ECM630 electroporator (Harvard Apparatus). TdT-positive pups were selected and used to dissect the LWs to examine the expression of Anks1a-VN. For X-gal staining, LWs were washed with 1% deoxycholate and 10% NP40 in PBS and fixed with 1% PFA, 0.2% glutaraldehyde, 1% deoxycholate, and 10% NP40 in PBS for 10 min. The LWs were washed 3 times for 10 min and exposed to staining buffer (K$_4$Fe(CN)$_6$, K$_3$Fe(CN)$_6$, MgCl$_2$, X-gal) at 37 °C. After 15 h, the reaction was stopped by washing three times with PBS. To set up ependymal cell culture, LWs were dissected directly from the live mice at P0–P2, collected in Hank's solution, and washed once. After the LWs were incubated with 500 μl trypsin-EDTA for 2 min at 37 °C, the dissociated cells were re-suspended with 1× Dulbecco's modified Eagle's medium containing 10% fetal bovine serum, collected by centrifugation at 350 × g for 5 min, and seeded on a 24-well plate coated with poly-L-lysine and laminin. Cells at 90% confluence were further seeded onto coated-bottom dishes and induced into ependymal differentiation using serum-free medium.

**Fluorescence microscopy**. Data were collected using an LSM700 (Carl Zeiss Microscopy) with a Plan-Apochromat ×63/1.46 oil immersion objective lens of an AxioObserver camera. Images were taken at 0.2–0.5 μm z-stack intervals over a 3–5-μm thickness. The fluorophores were excited with 488-, 555-, and 639-nm laser wavelengths. All images were processed by the ZEN Black software. For 3DSIM analysis, data were collected using an ELYRA PS.1 (Carl Zeiss Microscopy) with an alpha Plan-Apochromat ×100/1.46 oil immersion objective lens of an additional ×1.6 optovar. An Andor AxioObserver EMCCD camera was used to acquire images with 50 nm/slice z-stack intervals over a 3–3.5-μm thickness. The fluorophores were excited with 488-, 561-, and 642-nm wavelengths. Band-pass 495–575, 570–650, long-pass 655 and 750 nm filters were used to collect the emission wavelengths. Laser powered at the objective focal plane of 52 mW in the 0.9–2.6% range, exposure time of 50 ms, and EMCCD camera gain value of 20 was used during image acquisition. For the image field, grid excitation patterns were collected for 5 phases and 3 rotation angles (−75.25°, −14.28°, +43.36°). The raw data were reconstructed using the SIM module of ZEN Black 3.0 software with noise filter values between −4.75 and −2.99. Following acquisition, images were imported into IMARIS 8.0.2 (Bitplane) with a background subtraction algorithm. The threshold for 3D analysis was applied following the automatic setting using the 0.01 μm surface setting and noise filter, eliminating structures of <0.01 μm$^3$.

**Linear 3DSIM measurement analysis**. To measure the axial distance between FOP and other BB proteins, 3DSIM images of FOP and other BB proteins on the same z-slice were analyzed. The intensity profiles were generated using Create Image built into the Zeiss Zen Blue software and the relative distance between the peak maxima of FOP and other BB proteins was determined using the profile function built into the Zeiss Zen Blue software (Supplementary Fig. 1d, top panel). For the radial distance measurement, maximum intensity projection was used to draw a full circle along the outside of the FOP ring image and then two peak maxima at the opposite sides of the ring image were used to determine a center (set to zero). Other circles containing each BB protein were drawn around the center and then the distance between the peak maxima for each BB protein and the center on the same plane was determined using the profile function (Supplementary Fig. 1d, bottom panel).

**Immunoblots and immunoprecipitation**. LWs from the mice at P4 were lysed in PLC lysis buffer (50 mM HEPES pH 7.5, 150 mM NaCl, 10% glycerol, 1% Triton X-100, 1.5 mM MgCl2, 1 mM EGTA, 10 mM NaPPi, 100 mM NaF) with protease inhibitors (Roche) on ice. Protein extracts were collected by centrifugation at $14,000 \times g$ for 10 min. To perform immunoprecipitation[29,30], LW lysates were incubated with primary antibody (1 μg) for 90 min and with Protein-A Sepharose beads (GE Healthcare) for 30 min. The beads were washed with HNTG buffer (20 mM HEPES pH 7.5, 150 mM NaCl, 10% glycerol, 0.1% Triton X-100). Protein samples were separated by 10% sodium dodecyl sulfate-polyacrylamide gel electrophoresis, transferred onto a nitrocellulose membrane (Whatman), and incubated with primary antibodies diluted in 5% milk in TNTX (50 mM Tris, pH 7.6, 150 mM NaCl, 0.1% Triton X-100). The following antibodies were used: rabbit IgG anti-Anks1a (1:1000, Bethyl, A303-050A), mouse IgG2b anti-FOP (1:1000, Abnova, H00011116-M01), rabbit anti-CEP19 (1:1000, Proteintech, 26036-1-AP), and rabbit anti-CEP350 (1:1000, Novus, NB100-59811). The membranes were washed, blocked, and incubated with species-specific horseradish peroxidase-conjugated secondary antibodies for analysis using ECL Plus Western Blotting Substrate reagent (Thermo Scientific Pierce).

**Bead flow assay**. For the bead flow assay, LWs were freshly dissected and placed in L-15 media (Leibovitz) at 37 °C. Then a glass micropipette filled with Fluo-Spheres microbeads (2 μm, Thermo Fisher) attached to CellTram 4r Oil micromanipulator (Eppendorf) was lowered onto the LWs, where microbeads were deposited onto the ventricular surface. We recorded the movement of microbeads using ZEISS Axio Zoom.V16 fluorescent dissection microscope and ZEISS Axiocam 503 mono camera plugged into the ZEN imaging software (ZEISS) at 1.4 frames per second (fps).

**Analysis of rotational polarity**. For rotational polarity analysis[6,15], the vectors were drawn using the Fiji software to calculate each BF angle and obtain a mean value of BF angles within a cell. The mean of each BF angle was normalized to 225°, a hypothetical BF angle consistent with the direction of CSF flow in the AD region, and the distribution of the normalized BF angles was plotted on a histogram using SigmaPlot. The percentage of vectors were plotted on a histogram in 20° bins. For tissue-wide rotational polarity analysis, the rotational angles of each BF were averaged for each cell, and the mean value was obtained from approximately 250 cells. Each mean value was normalized to 225° and plotted on a histogram.

**Electron microscopy**. For SEM, LWs were dissected from mouse brains and fixed with 2% PFA and 2.5% glutaraldehyde in 0.1 M phosphate buffer (pH 7.4) for overnight at 4 °C. The LWs were rinsed with 0.1 M phosphate buffer (pH 7.4) three times, washing out any remaining aldehydes with 10% sucrose solution. The LWs were then incubated with 1% OsO₄ for 1 h on ice. After post-fixation, the LWs were washed with distilled water several times and dehydrated by immersion in graded concentrations of ethanol (65, 75, 85, 95, 99, and 100%) for 15 min each. The dehydrated LWs were placed on a dish with isoamyl acetate for 10 min; the isoamyl acetate was then removed, and the LWs were transferred to new dishes. The LWs were incubated at −80 °C deep-freezer to dry out overnight. The LWs were then mounted on metal stubs and were coated using a magnetron sputter coater. Samples were analyzed using a JSM-7600F (JEOL), field emission SEM. For TEM, brains were sectioned in coronal planes and 200-μm sections using a vibratome. Sections were fixed in 2.5% glutaraldehyde and 2% PFA solution overnight at 4 °C. Sections were washed with 0.2 M sodium cacodylate for 10 min for three times and treated with 2% OsO₄ for 1 h. The post-fixed sections were washed with distilled water several times and incubated in 1% uranyl acetate for overnight at 4 °C. The sections were washed with distilled water for 10 min three times and progressively dehydrated in 20, 50, 70, 80, 90, 95, and 100% ethanol. Sections were transited with ethanol/epon mixture (3:1, 1:1, 1:3) for 1 h each and finally impregnated into epon for overnight. The sections were mounted in epon blocks for 48 h at 60 °C oven. Mounted blocks were sectioned using an ultramicrotome (MTX, RMC) and ultrathin sections (50–70 nm) were post-stained by 2% uranyl acetate and 0.3% lead citrate. Samples were analyzed using JEM1010 (JEOL) TEM. For analysis of MTs and intermediate filaments, 50-nm-thick ultrathin serial sections (10–15 sections) were placed on a one-hole grid and analyzed by TEM (Talos L120C). Each image was manually adjusted in Adobe Photoshop to align each image into a 3D array. Then MTs and intermediate filaments in each sections were drawn as green and orange lines, respectively. In particular, we analyzed MTs and intermediate filaments present in the 450-nm (9 sections) region beneath the section showing the initial portion of BF. Newly generated TIFF images were imported into the Zeiss Zen Blue software to reconstruct a 3D image. In addition, each TIFF image was also imported into the AngioTool software to quantify the length of MTs and intermediate filaments.

**Quantification and statistics**. Statistical analysis was conducted on data from three biologically independent experimental replicates. Error bars displayed on graphs represent the mean ± SD of three independent experiments. Statistical significance was analyzed using unpaired Student's *t* test for two groups in Figs. 2d, f, g, j, k; 3g; 4c, e; 5d, g, i; 6c, d, e, h, i, k; 7b, d, g, j, h, k; 8b, d and Supplementary Figs. 2b, d, f, g; 3b, d, f; 4g; 5b, f; 7f, g, h, i or ordinary one-way analysis of variance

test for multiple groups in Fig. 1d. All dot plot analyses were performed using GraphPad Prism 5.01. Histogram analysis were performed using SigmaPlot 7.0.

**Reporting summary**. Further information on research design is available in the Nature Research Reporting Summary linked to this article.

## Data availability
The data that support the findings of this study are available from the corresponding author upon reasonable request. Raw microscopic image data are deposited to https://doi.org/10.8888/EPOPS20201221117. Source data are provided with this paper.

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

## Acknowledgements

This work was supported by grants NRF-2018R1A2B5A02021738 and NRF-2018M3C7A1056276 from the National Research Foundation of Korea (NRF).

## Author contributions

S.P. supervised the study. H.R. performed LW dissection, immunohistochemistry, and analysis of rotational and translational polarity. H.L. performed immunoprecipitation, linear 3DSIM measurement analysis, and electron microscopic analysis. H.R. and H.L. contributed to the interpretation of all experiments. H.R., H.L., J.L., M.S., and S.H. performed statistical analyses. J.L. and M.S. contributed to prepare mice samples. H.N. contributed to the production of mice lines and performance of bead flow assay. V.K. and J.K. contributed to the performance of *Xenopus* study. S.H. performed real-time quantitative RT-PCR analysis. S.P., H.R., and H.L. wrote the manuscript.

## Competing interests

The authors declare no competing interests.
