## [Peer Review File · Nature Communications]

REVIEWER COMMENTS

Reviewer #1 (Remarks to the Author):

Summary:

Ryu and colleagues have explored functional and structural consequences of Anks1a knockout on development, maturation and maintenance of the basal feet and motile cilia of ependymal multiciliated cells. A uniform alignment of the basal feet in these cells is a prerequisite for coordinated ciliary beating and proper flow of the fluid over the epithelial membrane. Such fluid delivers growth factors and other needed components and removes harmful metabolic products. How polarized subdistal appendages (SDA)/basal feet assemble in multiciliated cells is still unknown. Ryu and colleagues suggest that Anks1a, in association with subdistal protein FOP, assembles the inner portion of SDAs (region I). Removal of Anks1a disrupts the region I of SDAs. In turn, SDAs are destabilized and disoriented, and rotational polarity and the coordinated ciliary beating is perturbed. After Anks1a removal, a disorganization, in addition to the loss of ependymal SDAs and cilia appears even more pronounced in the brain of aged mice, which also sometimes showed enlarged brain ventricle.

General assessment: Imaging throughout the manuscript is generally good, and the data is well presented. The manuscript is clearly written. The effect of Anks1a knockout on tissue organization, fluid flow and SDA orientation are thoroughly analyzed. However, it remains speculative how Anks1a removal leads to the observed defects in SDAs and why the effects of Anks1a are more pronounced in ependymal cells of older brains. The manuscript could be suitable for the readership of *Nat. Commun.* if the concerns listed below are addressed.

Point 1. Biochemical data shows interaction between FOP and Anks1a. However, microscopy data shows only limited co-localization of these two proteins. Also, contrary to other SDA proteins, localization pattern of Anks1a at SDAs is not obvious (Fig. 1B and Sig S1 Anks1a-VN localization). Anks1a does not exclusively associate with the basal bodies.

Side projections which show all basal bodies in Figure S1D and E are not helpful because it is not clear which signal belong to which basal body. Why is Anks1a-VN signal in a different Z plane than FOP in Fig. 1SD? The cartoon in Fig. 1F is showing that it is in the same Z plane with FOP. How was Z shift determined? What was used for correction of the axial shift between different channels? Signals in blue are barely visible.

Finally, Line 360: the authors state that "Anks1a is frequently localized in transient puncta in small FOP-negative region". However, I don't clearly see such reproducible localization pattern. So, I feel that the conclusion that "Anks1a seems to play a role in carving the future region I from circular FOP negative regions" seems farfetched.

Point 2. In Figure 3B and. In KO cells, there is an obvious decrease in the levels of acetylated tubulin. This could signify that basal bodies in KO cells don't ciliate as efficiently as in wt cells. However, the authors do not mention this in the manuscript. The distribution of GT335 signal looks different as well. Higher resolution images should be included to clarify what is happening with cilia. This is significant because FOP has been previously implicated in early ciliation (ref #27 and #28). The density of basal bodies is also lower in *Xenopus* epidermis after administration of Anks1a morpholino oligonucleotides (Figure 7).

Point 3. How can authors be sure that SDA defects drive observed ciliary problems. Maybe basal body/ciliation defects lead to the observed SDA defects.

Point 4. The manuscript would be substantially stronger with high resolution ultrastructural TEM analyses of SDAs and basal feet in Anks1a control and KO cells are provided (figure 4D does not have sufficient resolution). Are SDAs in KO cells associated with cytoskeletal elements? Is the integrity of the basal bodies in Anks1a KO cells preserved, especially in the aged brain? Are all basal bodies properly docked to the cell surface in KO cells? The authors speculate that 'degeneration' of SDAs observed in the adult brain of KO cells could be a consequence of a more instable and fragile cytoskeletal network, but no analysis is done to explore that. It would be very important to show high-resolution ultrastructural data, especially because it remains elusive how Anks1a contributes to SDA structuring.

Point 5. FOP ciliogenesis and that association with Cep350, FOP was shown to be important for microtubule anchoring (PMID: 16314388) and its loss in human cells resulted in disorganization of microtubule networking how MT network looks in KO cells?

Point 4. In Line 84 the authors state that FOP is also localized to the basal feet/SDAs of primary cilia and reference ref #27, 28, and 29. Only reference #29 is showing localization of FOP to subdistal appendages.

Reviewer #2 (Remarks to the Author):

xAnks1a regulates the molecular dynamics of subdistal appendages in concert with FOP in multiciliated cells.

Hyunchul et al.

This is an interesting study unraveling some of the molecular mechanisms for how the initial polarity of ependymal cells (E1) basal bodies are established. Ciliary rotational polarity is essential for directional ciliary beating and the proper circulation of the cerebrospinal fluid. The work shows that Anks1a plays a key role, through FOP in the rotational polarization and alignment of E1 cell's cilia and that this protein is also essential for the stability of BB and cilia function in adults and during aging. The study shows that Anks1a in whole mounts of the LW of the LV is highly expressed during the first 2 weeks of postnatal development and at lower levels in the adult. Anks1a is present close to CNTRL in a ring structure formed by FOP in the SDAs. Immunoprecipitation shows that Anks1a binds to FOP and CEP350. Anks1 only partially localized to FOP (Basal rings) in immature E1 cells, but rarely seen in adults. CNTRL and ODF2 are localized to the FOP- arc in the basal rim and their localization moves from zone I to zone I and II as E1 cells mature. This FOP- arc decreased in Anks1a^{-/-} KO mice. They find that ODF2 levels are reduced in Zone I in Anks1a^{-/-} KO mice. Serial TEM analysis shows that the size of the BBs is not affected in Anks1a^{-/-} KO mice, but the base-line, height, and central angle were significantly different. Cilia are disorganized in the Anks1a^{-/-} KO mice and bead movement was significantly reduced; interestingly without hydrocephalus. Rotational polarity is significantly altered in Anks1a^{-/-} KO mice. Postnatal deletion of Anks1a also resulted in the abnormal ciliary organization, ciliary beating, and congested bead flow. Finally, the study shows that

Anks1a ortholog in *Xenopus* also has a role in the organization of BB in the epidermis and these effects were partially rescued by expression of mouse Anks1a. This is an important new set of findings that are well written and beautifully illustrated.

Major points:

The data are presented as N = number of cells or number of basal bodies, and the statistics are applied to this N. Would it be more appropriate to have the N represent the biological replicates (each mouse, or each culture)? There would be more confidence in the result if this were more clearly shown.

Minor points:

In Fig. 1 A, it would help to include in the figure the mice genotype.

It will be useful to specify what authors consider undifferentiated, immature, or Mature E1 cells;

Where these derived from specific ages?

It is unclear how the 67 BB were identified in immature E1 cells; if I understand correctly only 2 had FOP staining.

Reducing the number of abbreviations will make the reading of this paper easier.

As the authors introduce regions I and II, it would be helpful to include a diagram with the entire BB in Fig. 2 to orient the reader.

The TEM measured parameters are poorly illustrated in Fig. 2S. It will help to align their diagram in sFig. 2 D with the TEM images and include this diagram as part of fig 2. Measurements of the central angle and baseline are redundant if the ratios of the BB did not change.

The term PM in their flow analysis may be confusing, as some could think this refers to the 'posterior medial' wall. This region is also posterior to the adhesion region, but it's not part of the posterior LW.

It is puzzling why Anks1a retains important roles in the adult brain, but it is not detected in the SDA after P20. Does Anks1a have a different localization in the adult?

The straight Anks1a KO shows a dramatic reduction in cilia number and BB in aged mice. Does the conditional removal of Anks1a at P30 result in a similar phenotype? This could better establish a role for Anks1a in the stability of adult BBs.

Primary refs of the whole mount preparation should be provided.

Reviewer #3 (Remarks to the Author):

In this study, the authors characterized the assembly dynamics of the basal foot during ciliogenesis of motile cilia in ependymal cells. The authors developed reagents and assays to effectively document the process in which a mature, polarized structure seen at the centriole called the basal foot is developed or transformed from an unpolarized, radially symmetric structure called the subdistal appendages. Using these assays, the authors aimed to identify molecules regulating such a transformation, and found that Anks1a, a protein previously shown to have profound impacts on ependymal cell differentiation and development, also interacts with subdistal appendages and is

required for proper basal foot assembly and maintenance.

The assays/reagents that the authors developed to visualize the transformation of the basal foot from subdistal appendages are beautifully done and intriguing (Figure 2A & B). What they showed (Figure 2A&B) clearly indicates the existence of a complicated program that reshapes the subdistal appendages into a polarized basal foot. What is not clear to me, however, is whether or not Anks1a is directly involved in basal foot maturation (see below).

Major issues:

Anks1a is known to be essential for ependymal cell differentiation and development; it has also been shown that Anks1a is involved in COPII-mediated vesicle trafficking. The basal foot phenotypes seen in Anks1a^{-/-} animals or cells may thus likely be the secondary (non-specific) consequence of the impact resulted from loss of Anks1a's real activity in vesicle trafficking and/or ependymal cell differentiation. The authors certainly did not provide enough data to rule out this important possibility that would completely alter their conclusion.

For example:

a). The data in Fig 1B-H, which was used to link Anks1a with subdistal appendages, is unclear and poorly prepared. To me, from the images provided, the Anks1a punta seem randomly present around the area where basal bodies are located, only some of which happened to colocalize with FOP signals. As a result, in cases where Anks1a does colocalize with FOP, it is at a spot or position near the FOP ring that is not known to correspond to any elements or structure of the centriole that would make that "spot" special. The authors need to conduct 3D localization studies with detailed quantifications to show where and when Anks1a starts to associate with FOP (depicting what the pattern is), and explain how the pattern correlates with the transformation of the basal foot in region 1 or 2 during maturation.

In addition, the CO-IP result in Fig 1G looks strange to me – why would Anks1a antibodies proportionally pull down much more FOP than pulling down itself. Does it mean that the majority of FOP in cells is associated with Anks1a (in high stoichiometric ratio)? If so, why didn't we see more FOP/Anks1a co-localization in cells in any of images provided?

b). The loss-of-Anks1a phenotypes shown in mice, either at the cellular/subcellular level (Fig2 and 4) or system level (Fig 3, 5 & 6), can all be explained by the possibility that Anks1a^{-/-} cells may simply fail to maintain intact cell polarity including the planar cell polarity (PCP). For example, lack of PCP would explain the random orientation of the basal foot in Anks1a^{-/-} cells. Given the fact that Anks1a has profound impacts on ependymal cell differentiation and development, the authors need to show that the basal foot or cilia defect reported in the study is not the secondary consequence of Anks1a^{-/-} cells losing their intact cell polarity.

c). Similarly, the random orientation of the basal foot seen in *Xenopus* embryos depleted of Anks1a (Fig. 7A & C) also appears to correlate with a defect in centriole clustering – i.e. centrioles in cells depleted of Anks1a are more apart from each other, showing lower density per surface area, in addition to being mis-oriented. This again suggests a defect in cell polarity in *Xenopus* when Anks1a

is lost. A similar concern regarding centriole clustering/density is noted in Figure 4 as well. The authors need to provide detailed quantifications about centriole density in all experiments and whether cell polarity including PCP is affected in these cells.

Reviewer 1:

Imaging throughout the manuscript is generally good, and the data is well presented. The manuscript is clearly written. The effect of Anks1a knockout on tissue organization, fluid flow and SDA orientation are thoroughly analyzed. **However, it remains speculative how Anks1a removal leads to the observed defects in SDAs and why the effects of Anks1a are more pronounced in ependymal cells of older brains. The manuscript could be suitable for the readership of Nat. Commun. if the concerns listed below are addressed.**

Point 1. Biochemical data shows interaction between FOP and Anks1a. However, microscopy data shows only limited co-localization of these two proteins. Also, contrary to other SDA proteins, localization pattern of Anks1a at SDAs is not obvious (Fig. 1B and Fig S1 Anks1a-VN localization). Anks1a does not exclusively associate with the basal bodies. Side projections which show all basal bodies in Figure S1D and E are not helpful because it is not clear which signal belong to which basal body. Why is Anks1a-VN signal in a different Z plane than FOP in Fig. S1D? The cartoon in Fig. 1F is showing that it is in the same Z plane with FOP. How was Z shift determined? What was used for correction of the axial shift between different channels? Signals in blue are barely visible. Finally, Line 360: the authors state that “Anks1a is frequently localized in transient puncta in small FOP-negative region”. However, I don't clearly see such reproducible localization pattern. So, I feel that the conclusion that “Anks1a seems to play a role in carving the future region I from circular FOP negative regions” seems farfetched.

Rebuttal

The reviewer raises a major point that the co-localization study of Anks1a with FOP in SDAs is of poor quality. We feel that Anks1a antibodies are not of high quality as compared with FOP and other BB markers. Another problem is that Anks1a protein is not abundantly detectable in ependymal cells for an unknown reason and that they are only found as puncta. The very restricted and transient localization of Anks1a does contrast with FOP which is abundant and forms a large ring-like structure. In this respect, the ectopic expression of Anks1a-VN was critical in confirming the co-localization of endogenous Anks1a with FOP.

The data presented in Fig. 1C-E were generated using a linear 3DSIM analysis for various BB proteins from immature cells. Please see the detailed description for this method in the Experimental Methods (page 27, line21-25; page 28, lines 1-20). Briefly, the 3D-SIM, three-channel, z-stack images of protein-of-interests were taken in 488, 561 and 642 nm channels, and then reconstructed and aligned in the SIM module of ZEN Black 3.0 software. For each triplet of the images, the intensity profiles were generated using Create Image, and the peaks brighter than 30% of the maximum intensity were fit with a one-dimensional Gaussian function. For linear measurement, we first collected many FOP-positive ring images (n=63) to avoid distortions due to anisotropic resolution and then selected these images as a reference for each experiment (Fig. 1C, second panels). For the axial distance measure, 3D SIM images of FOP and other BB proteins on the same z-slice were analyzed for the measure of the relative distance between the peak maxima of FOP and other BB proteins (sFig. 1D, top panel). Note that the maximal intensity point of FOP signal was set to zero and then those of other BB proteins along the same axial direction were presented (Fig. 1D). For the radial distance measurement, maximum intensity projection was used to draw a full circle along the outside of each FOP ring image and then two peak maxima at the opposite sides of the ring image were used to determine a center (set to zero). Other circles containing each BB protein were drawn around the center and then the distance between the peak maxima for each BB protein and the center on the same plane was determined using the profile function (sFig. 1D, bottom panel). The cartoon in Fig. 1F is based on these experimental data.

To show a better and improved co-localization of Anks1a with FOP, we adopted the Imaris software to generate three dimensional FOP image renderings with Anks1a depicted in red color (Fig. 1B). The same tool was also utilized to generate 3D FOP image renderings with the ectopic localization of Anks1a-VN in red color (Fig. 1H-J). Our improved image analysis shows that 74% of double positive FOP⁺CNTRL⁺ SDAs were co-localized with Anks1a-VN in immature ependymal cells (sFig. 1F and G). However, Anks1a was detectable only as puncta in close proximity to the ring-shaped FOP (Fig. 1I). Consistent with this analysis, anti-Anks1a antibodies proportionally pulled down much more FOP than pulling down Anks1a (Fig. 1G). One reason may be that FOP contains the LisH domain, critical for dimerization and centrosomal localization and in addition, LisH domains in other proteins such as Muskelein have been shown to be critical for oligomerization (please see references 39 and 40). Importantly, our bimolecular fluorescent complementation analysis showed that two or more FOP proteins associate with each other in ependymal cells (sFig. 1E). Therefore, it is likely that the FOP protein complex co-precipitated with Anks1a containing a much higher stoichiometric ratio of FOP. We think that this biochemical result is consistent with the very limited localization of Anks1a detected in the FOP-stained SDAs (Please see the revised manuscript, page 7, lines 1-5; page 18, lines 10-14).

The reviewer also states that “Anks1a plays a role in carving the future region I from circular FOP negative regions” is a farfetched conclusion. The data presented in Fig. 1K were based on our statistical analysis of several 3D SIM images. For example, 74% of the double-positive FOP⁺CNTRL⁺ SDAs were co-localized with Anks1a puncta, while 60% of the triple-positive Anks1a⁺FOP⁺CNTRL⁺ SDAs containing Anks1a puncta localized to the FOP-negative regions (sFig. 1F and G). Based on these observations, we postulate, but not conclude, that Anks1a has a role in carving the future region I from circular FOP negative regions (please see the revised manuscript, page 7, lines 17-19; page 17, lines 8-10).

Point 2. In Figure 3B and. In KO cells, there is an obvious decrease in the levels of acetylated tubulin. This could signify that basal bodies in KO cells don't ciliate as efficiently as in wt cells. However, the authors do not mention this in the manuscript. The distribution of GT335 signal looks different as well. Higher resolution images should be included to clarify what is happening with cilia. This is significant because FOP has been previously implicated in early ciliation (ref #27 and #28). The density of basal bodies is also lower in *Xenopus* epidermis after administration of Anks1a morpholino oligonucleotides (Figure 7).

Rebuttal

The reviewer raises a concern that basal bodies in Anks1a KO cells do not ciliate as efficiently as in wild type cells. A previous study showed that acetylated tubulin is more abundantly present in the tip of cilia while glutamylated tubulin is in the base of cilia (please see reference 36). Therefore, it is very likely that a decrease in the levels of acetylated tubulin in Anks1a KO cells results from a lack of proper bending or bundling of the ciliary tips. These set of disorganized ciliary tips are likely due to uncoordinated ciliary beating resulting from a rotational planar polarity defect. Importantly, our scanning electron microscopic (SEM) study also demonstrated that the overall growth of multi-cilia was not significantly impaired in young adult KO mice (Fig. 3D). Therefore, our data indicate that the basal bodies in KO cells do ciliate as efficiently as in wild type cells at least until mice reach two months of age (please see the revised manuscript, page 9, lines 16-18).

Point 3. How can authors be sure that SDA defects drive observed ciliary problems. Maybe basal body/ciliation defects lead to the observed SDA defects.

Point 4. The manuscript would be substantially stronger with high resolution ultrastructural TEM analyses of SDAs and basal feet in Anks1a control and KO cells are provided (figure 4D does not have sufficient resolution). Are SDAs in KO cells associated with cytoskeletal elements? Is the integrity of the basal bodies in Anks1a KO cells preserved, especially in the aged brain? Are all basal bodies properly docked to the cell surface in KO cells? The authors speculate that 'degeneration' of SDAs observed in the adult brain of KO cells could be a consequence of a more instable and fragile cytoskeletal network, but no analysis is done to explore that. It would be very important to show high-resolution ultrastructural data, especially because it remains elusive how Anks1a contributes to SDA structuring.

Point 5. FOP ciliogenesis and that association with Cep350, FOP was shown to be important for microtubule anchoring (PMID: 16314388) and its loss in human cells resulted in disorganization of microtubule networking how MT network looks in KO cells?

Rebuttal (points 3, 4, and 5)

The reviewer raises a strong point that basal body defects also lead to the observed SDA defects. To address this, we first used CEP164 antibodies to examine the BBs of two month old KO mice. CEP164 antibody is an excellent marker for distal appendages of mature BBs but also for analyzing their proper docking to the apical cell surface. This analysis revealed that the number of BBs per cell and also their density in each patch were not significantly altered in KO ependymal cells as compared with wild type cells (sFig. 2E and F). In addition, all basal bodies in KO cells were properly docked to the apical surfaces just like those in wild type cells (sFig. 2E). These results strongly suggest that the BB defect of Anks1a KO ependymal cells is restricted to the SDAs whereas distal appendages, apical docking and density are normal until KO mice become two months of age (please see the revised manuscript, page 8, lines 20-22). However, unlike Anks1a KO mice, knock-down of Anks1a in *Xenopus* results in a more severe BB defect such as having a more reduced density of basal bodies (sFig. 7F-I). A potential explanation underlying the differential BB defects between mouse and *Xenopus* could be the presence of a full-length Anks1b in mice as shown in sFig. 7A-D. Our data show that Anks1b is expressed in ependymal multi-ciliated cells in mice, raising a possibility that it has a functional redundancy with respect to Anks1a in maturation of basal bodies (sFig. 7J). Importantly, full-length Anks1b is absent in the *Xenopus* (sFig. 7C and D). This could be the reason why knock-down of Anks1a alone is sufficient in influencing the density of BBs in *Xenopus* epidermis, whereas this developmental defect in maturation of basal bodies was not observed in Anks1a KO mice (sFig. 4A-C) (please the revised manuscript, page 19, lines 16-25).

We agree with the reviewer that a high resolution ultrastructural TEM analyses will be helpful for visualizing SDAs, basal feet and cytoskeletal elements. There were some technical difficulties in obtaining high resolution TEM images for the LW tissues. However, we did our best to use serial TEM analysis to observe the basal bodies and microtubules (MTs). We scrutinized approximately 100 basal bodies from WT or KO samples (1.5 month-old littermates), indicating that the 9+2 ciliary MT structure, distal appendage and triplet MT structure were not disturbed in KO (sFig. 2H). In addition, we analyzed MT networks around BF with 3-D images presented, and we could not find any differences between MT networks of WT and KO samples (Fig. 7; sFig. 6). However, we did find that disorganization of MT networks was severe in aged KO mice, in particular, in regions where BBs were defective in the aged KO tissues (sFig. 7C and 7I). We also used alpha-tubulin staining to confirm these results. Please see the revised manuscript, page 8, lines 26-27; page 9, lines 1-2; page 14, lines 15-21.

Point 6. In Line 84 the authors state that FOP is also localized to the basal feet/SDAs of primary cilia and reference ref #27, 28, and 29. Only reference #29 is showing localization of FOP to subdistal appendages.

Rebuttal

Per the reviewer suggestion, only reference #27 is now used in our revised manuscript to cite a report regarding a specific localization of FOP to subdistal appendages (please see the revised manuscript, page 4, lines 22-23).

Reviewer #2:

This is an important new set of findings that are well written and beautifully illustrated.

Major points:

The data are presented as N = number of cells or number of basal bodies, and the statistics are applied to this N. Would it be more appropriate to have the N represent the biological replicates (each mouse, or each culture)? There would be more confident in the result if this were more clearly shown.

Rebuttal

The reviewer asks for distinguishing the biological replicates (N, the number of mice or cells) from the number of samples used (n, the number of basal bodies). We agree with the reviewer. In the figure legends of the revised manuscript, we designate N and n for the independent biological replicates and the number of samples such as basal bodies or other cellular structures, respectively (please see the revised manuscript, page 34, lines 14 and 16).

Minor points:

1. In Fig. 1 A, it would help to include in the figure the mice genotype. It will be useful to specify what authors consider undifferentiated, immature, or Mature E1 cells; Where these derived from specific ages?

Rebuttal

As the reviewer suggests, we have added the mice genotype to Fig. 1A. For analyzing the differential stages of E1 cells, the details are explained in the legend for sFig. 1C but also in the text. Briefly, we analyzed the LWs derived from mice at P4 (undifferentiated and immature cells) and P21 (mature cells). Undifferentiated cells contain two centrioles whereas immature or mature cells have multiple centrioles. In immature cells, most of centrioles were localized to the apical surface but most of the apically docked BBs displayed a random orientation in the FOP-negative regions. In contrast, in mature cells, almost all of apically docked BBs display a uniform alignment in the FOP-negative regions. We used these criteria to quantify the data in Fig. 1H-K (please see the revised manuscript, page 7, lines 12-21).

2. It is unclear how the 67 BB were identified in immature E1 cells; if I understand correctly only 2 had FOP staining.

Rebuttal

Per the reviewer suggestion, we have modified the Fig. 1K legend: n = 67 (the number of undifferentiated cells used for this analysis was 134), meaning that we analyzed 134 undifferentiated cells to assess 67 BBs positive for both FOP and CNTRL staining (please see the revised manuscript, page 35, lines 2-3).

3. Reducing the number of abbreviations will make the reading of this paper easier.

Rebuttal

As the reviewer suggests, we no longer use abbreviations such as UD, IM and M in the revised text (please see page 7, lines 12-21).

4. As the authors introduce regions I and II, it would be helpful to include a diagram with the entire BB in Fig. 2 to orient the reader.

Rebuttal

In Fig. 2B, we have now included a diagram depicting the apical and basal parts of the entire BB.

5. The TEM measured parameters are poorly illustrated in Fig. 2S. It will help to align their diagram in sFig. 2 D with the TEM images and include this diagram as part of Fig 2. Are measurements of the central angle and baseline redundant if the ratios of the BB did not change.

Rebuttal

We agree that the provided central angle and baseline were redundant. In the revised manuscript, a diagram for the BF parameters (Fig. 2I) is shown next to the serial TEM images (Fig. 1H). In addition, the baseline parameter is removed from Fig. 2 and the central angle data for BF are shown in Fig. 1K.

6. The term PM in their flow analysis may be confusing, as some could think this refers to the 'posterior medial' wall. This region is also posterior to the adhesion region, but it's not part of the posterior LW.

Rebuttal

Although the term PM may be confusing, this term was first introduced by another research group (reference #6) to describe a directed flow of CSF around the adhesion area of LW. We have used the same term to demonstrate that the directed flow of CSF is reproducibly observed in WT but not in Anks1a KO LWs. In addition, the same term is used to compare the ciliary growth pattern in the LWs. In our study, it is important to compare anatomically similar region between WT and KO ependymal tissue using well-known landmarks such as the adhesion area around AD, AV and PM.

7. It is puzzling why Anks1a retains important roles in the adult brain, but it is not detected in the SDA after P20. Does Anks1a have a different localization in the adult? The straight Anks1a KO shows a dramatic reduction in cilia number and BB in aged mice. Does the conditional removal of Anks1a at P30 result in a similar phenotype? This could better establish a role for Anks1a in the stability of adult BBs.

Rebuttal

The reviewer asks whether Anks1a is expressed in the adult brain and its conditional ablation in the adult brain results in the same phenotype as we found in the conventional aged KO mice. To address this issue, we dissected out the LWs from Anks1a^{+/*lacZ*} mice at 3 months of age to examine the expression of Anks1a. As a result, we were able to confirm that Anks1a is expressed in ependymal multi-ciliated cells in the adult brain. This suggests to us that Anks1a is expressed in the adult ependymal cells although at a low level. To examine a role for Anks1a in the stability of adult BBs, we generated iKO mice by injecting TM into 3 month-old mice and then analyzed their brains at 20~22 months. As shown in sFig. 5E-G, both BBs and motile cilia were significantly decreased in the aged iKO mice. These results strongly support our hypothesis that Anks1a is critically involved in regulating the dynamics of SDAs even in fully mature cells (please see the revised manuscript, page 14, lines 9-14). Note that this experiment had been in progress before the first submission to *Nature Communications*.

8. Primary refs of the whole mount preparation should be provided.

Rebuttal

The references for the whole mount LW preparation are now provided in the revised manuscript (please see the revised manuscript, page 26, lines 9-10).

Reviewer #3:

The assays/reagents that the authors developed to visualize the transformation of the basal foot from subdistal appendages are beautifully done and intriguing (Figure 2A & B). What they showed (Figure 2A&B) clearly indicates the existence of a complicated program that reshapes the subdistal appendages into a polarized basal foot. **What is not clear to me, however, is whether or not Anks1a is directly involved in basal foot maturation (see below).**

Major issues:

Anks1a is known to be essential for ependymal cell differentiation and development; it has also been shown that Anks1a is involved in COPII-mediated vesicle trafficking. The basal foot phenotypes seen in Anks1a^{-/-} animals or cells may thus likely be the secondary (non-specific) consequence of the impact resulted from loss of Anks1a's real activity in vesicle trafficking and/or ependymal cell differentiation. The authors certainly did not provide enough data to rule out this important possibility that would completely alter their conclusion.

For example:

a). The data in Fig 1B-H, which was used to link Anks1a with subdistal appendages, is unclear and poorly prepared. To me, from the images provided, the Anks1a puncta seem randomly present around the area where basal bodies are located, only some of which happened to colocalize with FOP signals. As a result, in cases where Anks1a does colocalize with FOP, it is at a spot or position near the FOP ring that is not known to correspond to any elements or structure of the centriole that would make that "spot" special. The authors need to conduct 3D localization studies with detailed quantifications to show where and when Anks1a starts to associate with FOP (depicting what the pattern is), and explain how the pattern correlates with the transformation of the basal foot in region 1 or 2 during maturation. In addition, the CO-IP result in Fig 1G looks strange to me – why would Anks1a antibodies proportionally pull down much more FOP than pulling down itself. Does it mean that the majority of FOP in cells is associated with Anks1a (in high stoichiometric ratio)? If so, why didn't we see more FOP/Anks1a co-localization in cells in any of images provided?

Rebuttal

The reviewer raises a major concern that the basal foot phenotypes may be the secondary consequences of loss of Anks1a's real activity in vesicle trafficking and /or ependymal cell differentiation. The reviewer provides three examples for this concern.

We agree with the reviewer that the co-localization study of Anks1a with FOP in SDAs could be better demonstrated. We feel that Anks1a antibodies are not of high quality as compared with FOP or other BB markers. Another problem is that Anks1a protein is not abundantly detectable in ependymal cells for unknown reasons and that they are found only as puncta. The very restricted and transient localization of Anks1a does contrast with that for FOP which is abundant and forms a large ring-like structure. As such, the ectopic expression of Anks1a-VN was a critical tool for us for confirming the co-localization of endogenous Anks1a with FOP. To further demonstrate co-localization of Anks1a with FOP, we adopted the Imaris software to generate three dimensional FOP-image rendering with Anks1a depicted in red color (Fig. 1B). This tool was also used to generate 3D FOP image rendering with the ectopic localization of Anks1a-VN (also in red) (Fig. 1H-J). Our improved image analysis shows that 74% of double positive FOP⁺CNTRL⁺ SDAs were co-localized with Anks1a-VN in immature ependymal cells (sFig. 1F and G). However, Anks1a was detectable only as puncta in close proximity to the ring-shaped FOP (Fig. 1I). Consistent with this analysis, Anks1a antibodies proportionally pulled down much more FOP than pulling down Anks1a

(Fig. 1G). One reason may be that FOP contains the LisH domain, critical for dimerization and centrosomal localization and in addition, LisH domains in other proteins such as Muskelein have been shown to be critical for oligomerization. Importantly, our bimolecular fluorescent complementation analysis showed that two or more FOP proteins associate with each other in ependymal cells (sFig. 1E). Therefore, it is likely that the FOP protein complex co-precipitated with Anks1a containing a much higher stoichiometric ratio of FOP. We think that this biochemical result is consistent with the very limited localization of Anks1a detected in the FOP-stained SDAs (please see the revised manuscript, page 7, lines 1-5; page 18, lines 10-14).

b). The loss-of-Anks1a phenotypes shown in mice, either at the cellular/subcellular level (Fig2 and 4) or system level (Fig 3, 5 & 6), can all be explained by the possibility that Anks1a^{-/-} cells may simply fail to maintain intact cell polarity including the planar cell polarity (PCP). For example, lack of PCP would explain the random orientation of the basal foot in Anks1a^{-/-} cells. Given the fact that Anks1a has profound impacts on ependymal cell differentiation and development, the authors need to show that the basal foot or cilia defect reported in the study is not the secondary consequence of Anks1a^{-/-} cells losing their intact cell polarity.

Rebuttal

The reviewer is concerned that Anks1a KO ependymal cells may be failing to maintain intact PCP due to the developmental defects. We feel regretful that the reviewer did not consider the inducible knock-out (iKO) experiments shown in Fig. 5. In the revised manuscript, we also provide additional data regarding the aged iKO mice (sFig. 5D-G), where floxed Anks1a mice with Cre-ER were allowed to undergo normal development until they were one or three months of age. Then, tamoxifen treatment led to ablation of the floxed Anks1a gene in ependymal cells. These iKO mice had the basal foot phenotypes of young adult mice but had the more severe BB and cilia phenotypes of aged mice (please see the revised manuscript, page 13, lines 2-4; page 14, lines 9-14).

We further addressed the reviewer's concern with two additional experiments. First, we used qRT-PCR analysis to examine whether various marker genes related with ependymal differentiation and maturation were differentially expressed in Anks1a KO cells. As shown in sFig. 4A-C, we did not find any significant differences in gene expression of progenitor (Ki67), cell fate determination (GemC1 and Mcidas), centriole amplification (Deup1 and Cep152), cilia growth (FoxJ1), and planar cell polarity (PCP, Vangl1, Vangl2, Fzd3, Celsr1-3) markers between WT and KO samples. These results suggest that Anks1a loss did not affect the overall differentiation program of ependymal multiciliated cells. Second, we followed representative PCP proteins for visualizing their distinct localization along the apical plane (sFig. 4D). In our experiments using commercially available antibodies, we found that Fzd3 and Vangl1 were detectable in the whole-mount LW immunostainings. In wild type cells, Fzd3 was localized to the distal side of the cells, whereas Vangl1 was present in their proximal side. This asymmetric localization of FZD3 and Vangl1 was not disturbed in Anks1a KO ependymal cells, indicating that Anks1a KO cells maintained intact PCP. Taken together, these results rule out the possibility that Anks1a deficiency affects normal differentiation program of ependymal cells and that the BF phenotype seen is the secondary outcome (please see the revised manuscript, page 11, lines 14-23).

c). Similarly, the random orientation of the basal foot seen in *Xenopus* embryos depleted of Anks1a (Fig. 7A & C) also appears to correlate with a defect in centriole clustering – i.e. centrioles

in cells depleted of Anks1a are more apart from each other, showing lower density per surface area, in addition to being mis-oriented. This again suggests a defect in cell polarity in *Xenopus* when Anks1a is lost. A similar concern regarding centriole clustering/density is noted in Figure 4 as well. The authors need to provide detailed quantifications about centriole density in all experiments and whether cell polarity including PCP is affected in these cells.

Rebuttal

The reviewer raises the point that the lower density of BBs observed in Anks1a MO-injected *Xenopus* could be also be seen in mice, thereby leading to the random orientation of the basal foot. Since a similar concern was raised by the first reviewer, the same response is shown here. First, we used CEP164 antibodies to examine both apical docking and the proper number of BBs in two month-old WT and KO mice (sFig. 2E and F). It is well known that CEP164 antibodies are an excellent marker of distal appendages in mature BBs but also for analyzing their apical docking to cell surfaces. This analysis revealed that the number of BBs but also their density in each patch were not significantly altered in KO ependymal cells as compared with WT cells. In addition, all BBs in KO cells were properly docked to the apical surface just as in wild type cells (sFig. 2E). These results strongly indicate that the BB defect of Anks1a KO ependymal cells is restricted to the SDAs whereas the distal appendages, apical docking and density are normal until Anks1a KO mice become two months old. However, unlike Anks1a KO mice, knock-down of Anks1a in *Xenopus* resulted in more severe BB defects such as more reduced density of BBs (sFig. 7F-I). A potential explanation underlying the differential BB defects between mouse and *Xenopus* could be the presence of a full-length Anks1b in mice as shown in sFig. 7A-D. Our data show that Anks1b is expressed in ependymal multi-ciliated cells in mice, raising a possibility that it has a functional redundancy with respect to Anks1a in maturation of basal bodies (sFig. 7J). Importantly, full-length Anks1b is absent in the *Xenopus*. This could be the reason why knock-down of Anks1a alone is sufficient in influencing the density of BBs in *Xenopus* epidermis. Please see the revised manuscript, page 8, lines 20-22; page 19, lines 16-25.

REVIEWERS' COMMENTS

Reviewer #4 (Remarks to the Author):

This significantly revised manuscript provides further details supporting the authors' conclusions. All points raised in the previous critique have been addressed.

Minor comment: Could the authors comment on the CEP164+GT335+ double-positive cilia in the adult iKO mouse whole mounts? This localization of CEP164 is unexpected, and appears to affect only a subset of cilia, even within the same cell.

Reviewer #4 (Remarks to the Author):

This significantly revised manuscript provides further details supporting the authors' conclusions. All points raised in the previous critique have been addressed.

Minor comment: Could the authors comment on the CEP164⁺GT335⁺ double-positive cilia in the adult iKO mouse whole mounts? This localization of CEP164 is unexpected, and appears to affect only a subset of cilia, even within the same cell.

Rebuttal

The reviewer raises a point why ependymal cells in the aged iKO mice contain a subset of cilia positive for both CEP164 and GT335. We are sure that the staining of cilia with CEP164 is a nonspecific background resulting from the secondary antibody we used. Since the iKO cells have much more reduced number of basal bodies, we increased the intensity of CEP164 staining so that the background signal was also enhanced in the cilia. We observed that CEP164 staining was nonspecifically observed in the cilia of control cells when its intensity was increased. To eliminate a misinterpretation of the CEP164 localization to cilia, we replaced the previous figure with new one (please see sFig.5E).